# Taxonomy-guided selection of *Paraburkholderia busanensis* sp. nov.: a versatile biocontrol agent with mycophagy against *Colletotrichum scovillei* causing pepper anthracnose

Mohamed Mannaa,[1,2,3] Gil Han,[1,2] Taeho Jeong,[1,2] Minhee Kang,[1,2] Duyoung Lee,[1,2] Hyejung Jung,[1,2] Young-Su Seo[1,2]

**ABSTRACT**  This study introduces a novel approach based on the taxonomy-guided selection of bacterial biocontrol agents from a known beneficial taxonomic group. Following 16S rRNA screening, we focused on the genus *Paraburkholderia*, which harbors strains with large genomes and versatile benefits to plants. A strain designated P39 was selected, identified, and characterized for its biocontrol activity against *Colletotrichum scovillei*. Strain P39 exhibited antagonism against *C. scovillei* by producing compounds, including volatiles, with antifungal activity, both *in vitro* and on pepper fruits. Genomic, physiological, and biochemical analyses revealed that the selected strain represents a novel species, named *Paraburkholderia busanensis*. Genomic analyses provided insights into the fitness and biocontrol activities of the selected strains. Moreover, P39 displays mycophagy, consuming fungal mycelia and transforming them into bacterial biomass, particularly in nutrient-poor media supplemented with fungal mycelia. The genome harbored chitin and N-acetylglucosamine utilization genes, suggesting a proposed pathway for the utilization of fungal cells as a nutrient source. Microscopic observations further supported the ability of this strain to rupture and damage fungal hyphae, depriving them of their cellular constituents. This study successfully demonstrated the implementation of a taxonomy-guided approach for the selection of bacterial strains for biocontrol. These findings contribute to our understanding of biocontrol strategies, bacteria-fungi interactions, and the identification of *Paraburkholderia busanensis* sp. nov. as a potential candidate for the biocontrol of pepper anthracnose. Additionally, this strain serves as a valuable resource for antifungal compounds and volatiles, and for the study of bacteria-fungal interactions and mycophagy.

**IMPORTANCE**  Traditional control methods for postharvest diseases rely on fungicides, which cause human health and environmental concerns. This study introduces a taxonomy-guided strategy for selecting biocontrol agents. By focusing on *Paraburkholderia* group, which harbors diverse plant-beneficial strains, the inadvertent selection of harmful strains was circumvented, thereby obviating the need for laborious *in vitro* screening assays. A highly promising candidate, strain P39, has been identified, exhibiting remarkable biocontrol activity against *Colletotrichum scovillei*. Through comprehensive genomic, physiological, and biochemical analyses, P39 was characterized as a novel species within the *Paraburkholderia* genus and designated *Paraburkholderia busanensis*. Moreover, these findings deepen our understanding of bacterial-fungal interactions, as they elucidate a potential pathway for the utilization of fungal chitin, thereby enhancing our understanding of bacterial mycophagy. *P. busanensis* is a promising source of antifungal volatiles and putative novel secondary metabolites.

**KEYWORDS**  bacterial mycophagy, biocontrol, anthracnose, *Paraburkholderia*

Address correspondence to Young-Su Seo, yseo2011@pusan.ac.kr.

The authors declare no conflict of interest.

See the funding table on p. 25.

Postharvest diseases are widely recognized as the primary cause of significant losses in fruit and vegetable production, spanning various stages from harvesting to distribution and consumption (1). Among these, anthracnose is a severe disease that affects a range of fruits and vegetables, including pepper (*Capsicum annuum*), a major ingredient in many cuisines (2). This destructive disease poses a significant threat to pepper production, particularly during the postharvest phase (3). More than 30 species belonging to *Colletotrichum* were reported to cause anthracnose in pepper, and among them, *C. truncatum* and *C. scovillei* are the most prevalent causes of pepper anthracnose in Asia (4). *Colletotrichum scovillei*, the causative agent of pepper anthracnose, is a serious concern in Korea and has been reported in numerous locations worldwide (5, 6).

Although chemical control using pesticides is effective in managing postharvest diseases, there are restrictions on their use at this stage. These limitations stem from concerns regarding their impact on human health and the environment, the emergence of pesticide-resistant biotypes of postharvest pathogens, and the loss of registration of several fungicides, particularly during the postharvest stages (7). Consequently, the exploration of efficient alternatives has gained increasing importance as there is an urgent need to develop suitable alternatives to the harmful application of chemical fungicides. Substantial research efforts have been dedicated to reporting and proposing biocontrol agents with the potential to control postharvest diseases (8). Although these endeavors have not fully met expectations in terms of applicability, they have yielded promising successes, resulting in the availability of biocontrol-based products for managing postharvest diseases. Among the registered products based on biocontrol agents, Biosave, which utilizes *Pseudomonas syringae*, and Shemer, which utilizes *Metschnikowia fructicola*, are notable examples (7, 9).

Biological control during the postharvest stage presents a sustainable alternative to the use of harmful chemical pesticides, especially when treating consumable produce at this critical stage. The uniqueness of the postharvest system could offer the possibility to control environmental conditions in favor of the biocontrol agent (10). However, addressing biosafety concerns related to antagonists is essential when implementing biocontrol methods, particularly for postharvest diseases. Several mechanisms have been reported for biocontrol agents against postharvest pathogens, including competition for nutrients and space, mycoparasitism through the production of lytic enzymes and breakdown of the pathogen cell wall, production of antifungal metabolites and volatiles, induction of resistance, and biofilm formation (11). In addition to employing antagonists for postharvest applications, it is crucial to conduct research aimed at identifying biocontrol agents and candidate microbes that exhibit antagonistic activity against postharvest diseases. These agents have the potential to produce metabolites or volatiles that can be harnessed for disease control such as anthracnose (12).

In general, the screening process for the selection of biocontrol agents against plant pathogens traditionally involves the isolation of a large number of microbes that are subsequently subjected to *in vitro* screening methods. This is followed by the characterization and identification of the most effective agents. However, this conventional approach may inadvertently result in the selection of bioagents that share taxonomic affinities with harmful microbes or opportunistic human pathogens. The primary objective of this study was to develop a taxonomy-guided approach for identifying an efficient bacterial biocontrol agent from a pool of known beneficial microbial taxa. Furthermore, our study aimed to thoroughly characterize and identify the selected biocontrol agent and gain insights into its mechanism of action and interaction with the target pathogen. Finally, we aimed to propose an effective biocontrol agent with the potential to be utilized for the management of pepper anthracnose.

## RESULTS

### Taxonomy-guided selection of *Paraburkholderia* sp. nov. P39

The obtained partial sequences of the 16S rRNA from the 206 isolated bacteria were deposited in NCBI GenBank, and the accession numbers are listed in Table S1. Maximum-likelihood phylogenetic trees constructed from the 16S rRNA sequences of the 206 isolated bacterial strains in the presence of 16S rRNA from a reference *Paraburkholderia* strain (*P. azotifigens* NF2-5-3[T]) facilitated the selection of 33 closely related strains within *Burkholderia sensu lato* clade (Fig. 1A). *Paraburkholderia* includes strains known for their versatility and antagonistic activity against phytopathogenic fungi. As a result of the phylogenetic analysis, 33 bacterial strains that were closely clustered around the added *Paraburkholderia* reference sequence were selected and used for further testing.

The preliminary dual-culture screening assay of the 33 selected *Paraburkholderia*-related strains resulted in the selection of P39 based on its antagonistic activity against *C. scovillei* KC05 in the *in vitro* assay, which was represented as the maximum reduction in the mycelial growth area compared to the control and other tested strains (Fig. 1B).

### Antagonistic activity of the selected *Paraburkholderia* sp. nov. P39 against *C. scovillei* KC05 *in vitro*

In the dual-culture assay of PDA plates, treatment with the selected strain P39 resulted in a significant reduction in the *C. scovillei* KC05 mycelial growth as shown in the photographs and as evidenced in the evaluated mycelial growth area ($cm^2$) compared with the control plates (Fig. 2A and B). The co-cultivation of P39 with *C. scovillei* KC05 resulted in a significant reduction (85% reduction) in the fungal growth, represented as the mycelial dry weights/mL of the culture broth (Fig. 2C and D). Similarly, when the collected cell-free culture filtrates of strain P39 were added to PDB containing *C. scovillei* KC05, the mycelial growth was significantly reduced (49% reduction), as evidenced from the evaluated mycelial dry weight (Fig. 2E and F).

To test the influence of produced volatiles from P39, an *in vitro* test was performed on a bi-plate. The results indicated that volatiles produced by strain P39 resulted in a significant reduction in the mycelial growth of *C. scovillei* KC05, as shown in Fig. 3A, which was confirmed by the assessment of mycelial growth ($cm^2$) compared to the control plates (Fig. 3B). The volatiles of P39 were also found to influence the conidiation of *C. scovillei* KC05, as evidenced by the assessment of the total conidia produced by the fungal body and further confirmed by the assessment of the number of produced conidia/ $cm^2$, respectively (Fig. 3C).

Based on these results, the selected *Paraburkholderia* strain, P39, was confirmed to have antagonistic activity against *C. scovillei* KC05, with the potential to produce antifungal metabolites and volatiles.

### Biocontrol activity of the selected bacterial strain P39 against anthracnose by *C. scovillei* KC05 on pepper fruits *in vivo*

The *in vivo* assays were performed on healthy green peppers. The results indicated that dipping fresh pepper in bacterial suspension of P39 significantly reduced the development of typical anthracnose symptoms in fungal-inoculated pepper fruits 5 days after incubation, as evidenced in the photographs and the assessed anthracnose lesion diameters (Fig. 4A and B).

From the square plate assay, the volatiles produced by strain P39 also resulted in a significant reduction in the anthracnose symptoms caused by *C. scovillei* KC05 on pepper fruits, as evidenced by the photographs and the assessed anthracnose lesion diameters (Fig. 4C and D). These results confirmed the efficient biocontrol activity of strain P39 and the volatiles produced against anthracnose in the inoculated pepper fruits.

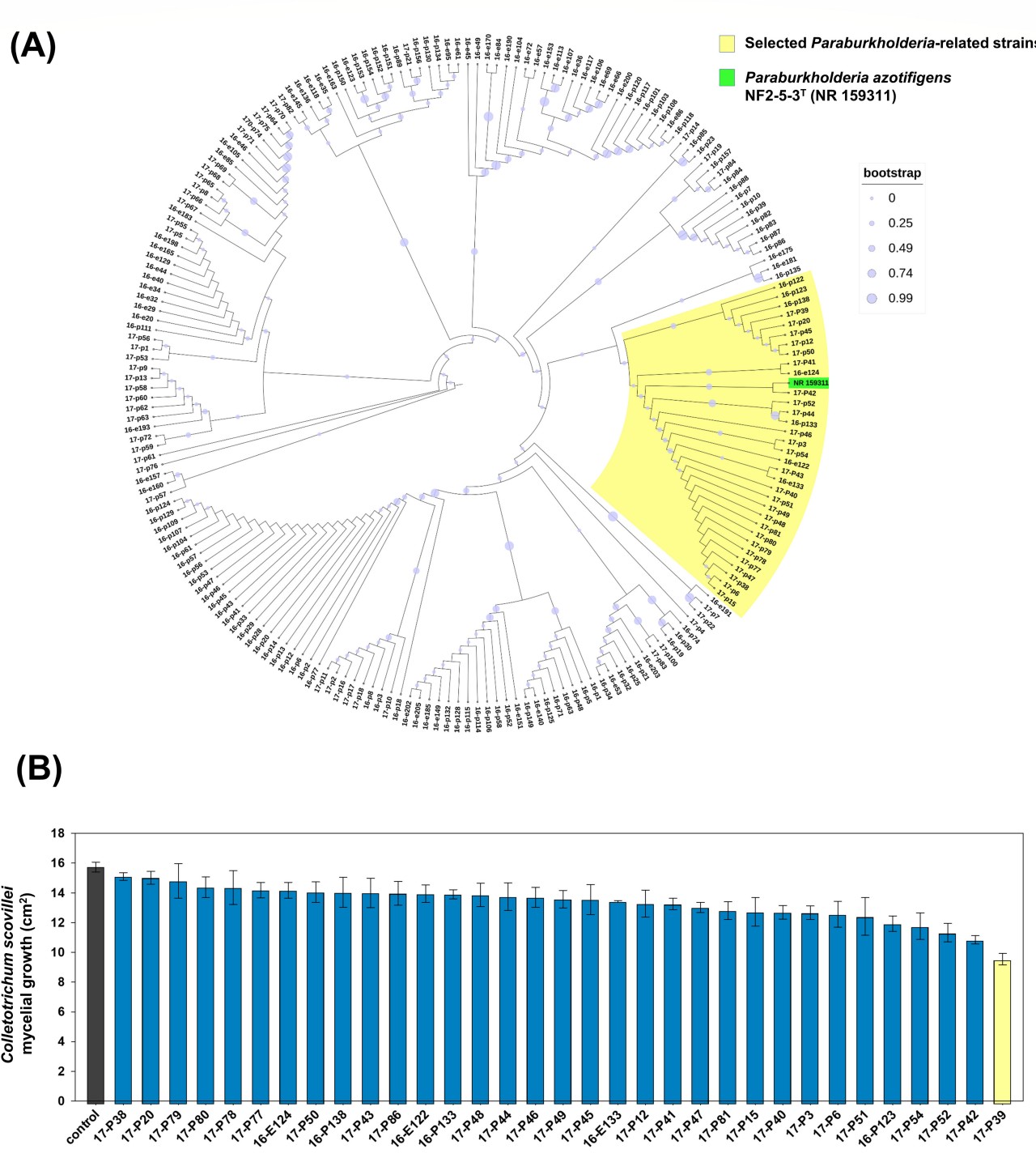

**FIG 1** (A) Phylogenetic tree constructed by the maximum-likelihood method from the partial 16S rRNA sequences obtained of 206 bacterial strains isolated from pine forests showing the taxonomic relationships among the isolated strains. *Paraburkholderia azotifigens* NF2-5-3T was included as a control to target the *Paraburkholderia*-related species within *Burkholderia sensu lato* group (highlighted in yellow) that were selected for biocontrol activity screening. (B) Bar graph representing the mycelial growth from *in vitro* dual culture assays of the selected 33 isolates of *Paraburkholderia*-related group for the selection of potential biocontrol bacterial strains against *Colletotrichum scovillei* KC05. Data presented are the means and standard errors for three replicates.

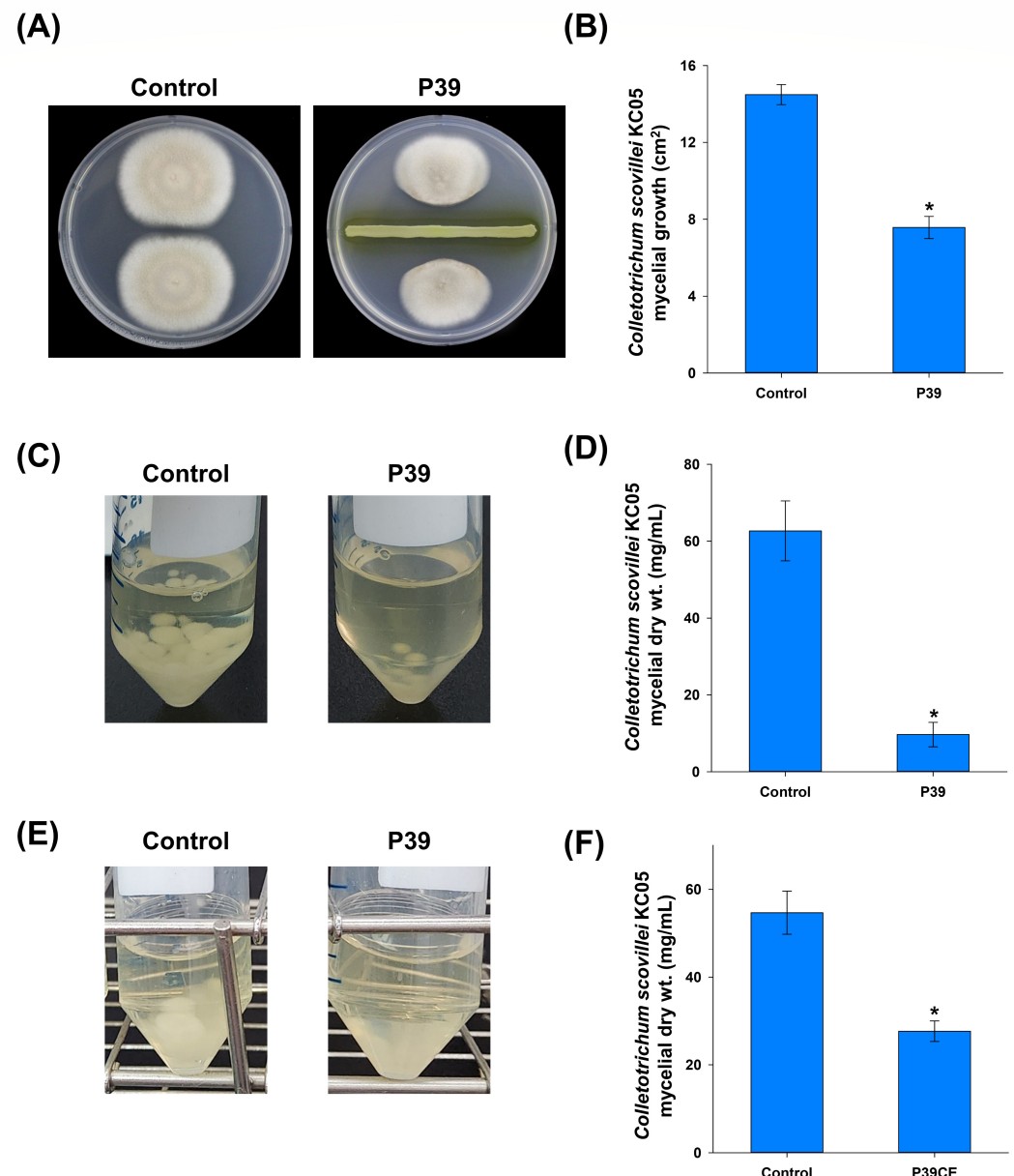

**FIG 2** *In vitro* antagonistic activity of the selected *Paraburkholderia* sp. nov. P39, against *Colletotrichum scovillei* KC05 in (A) dual culture on potato dextrose agar plates, (C) co-cultivation on potato dextrose broth (PDB), and (E) cell-free culture extract assay on PDB. The bar graphs on the right side of each assay represent the assessment of (B) the mycelial growth area, (D) and (F) the mycelial dry weight of *C. scovillei* KC05. The bar graphs were generated from the means and standard errors of three replicates, and the asterisks on the error bar represent significant differences according to a *t*-test at $P < 0.05$.

## Taxonomic identification of the bacterial strain P39 and comparative genomic characterization

The 16S rRNA sequence of P39 was compared with that of related type strains using NCBI BLAST. The results indicated similarity levels of 98.02%, 97.62%, 97.53%, 97.04, 96.83%, and 96.69% for *P. azotifigens* NF 2-5-3[T], *P. phymatum* LMG21445[T], *P. bryophila* LMG 23644[T], *P. fynbosensis* LMG 27177[T], *P. dilworthii* WSM3556[T], and *P. hospita* LMG20598[T], respectively. Neighbor-joining and maximum-likelihood phylogenetic analyses constructed using the 16S rRNA sequence of strain P39 and related *Paraburkholderia* strains resulted in the separation of P39 into a distinct cluster, which does not allow for its identification

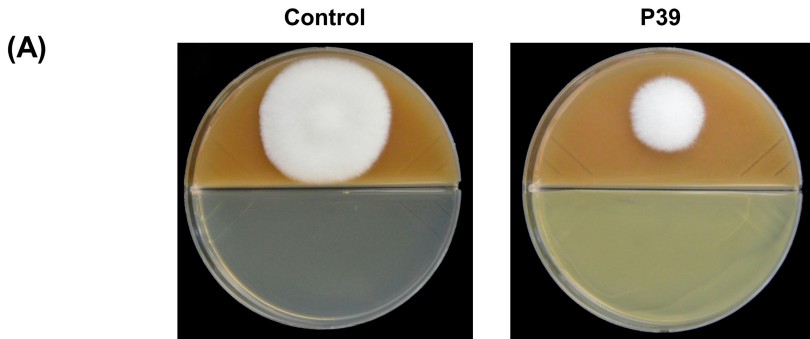

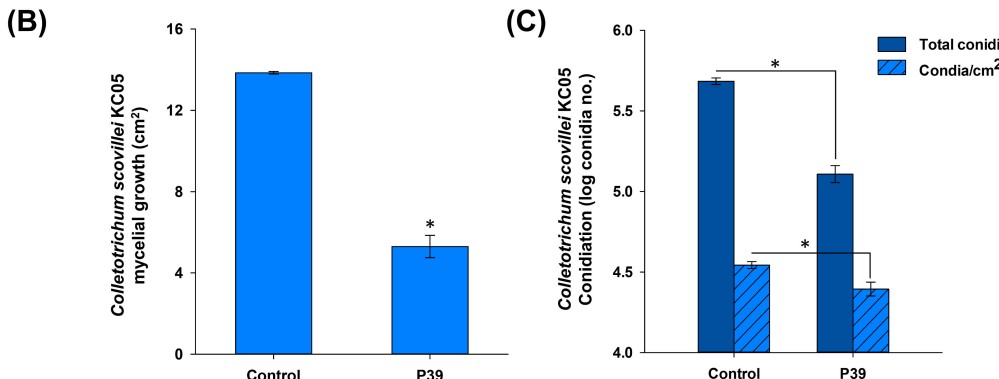

**FIG 3** Influence of volatiles produced by *Paraburkholderia* sp. nov. P39 on *Colletotrichum scovillei* KC05. (A) Bi-plate volatile assay where *C. scovillei* was grown on V8 juice agar and P39 grown on nutrient agar. (B) Bar graphs of the mycelial growth area in square centimeter, (C) influence of the volatiles produced by P39 on the conidiation of *C. scovillei* represented as total conidia from the bi-plate side of fungal growth and the produced conidia/cm$^2$ of the fungal mycelia. The asterisks on the error bar represent significant differences according to a *t*-test at $P < 0.05$.

at the species level (Fig. 5A). Therefore, P39 may be a novel species in the *Paraburkholderia* genus.

Based on these results, the complete genome sequence of P39 was compared with the genomes of related *Paraburkholderia* strains retrieved from the NCBI database. The details of the genomes used for comparison, including their accession numbers and basic features, are shown in Table 1. The *in silico* analyses of the P39 genome compared with those of the *Paraburkholderia* species were found to be below the cutoff value for species identification. Both the average nucleotide identity (ANI) and the digital DNA-DNA hybridization (dDDH) parameters were below the cutoff values of >95% and 70%, respectively, for species identification (Fig. 5B and C).

Furthermore, a phylogenomic tree was constructed using genome blast distance phylogeny (GBDP) based on the whole-genome sequences of strain P39 and related species within the *Paraburkholderia* genus. The resulting tree revealed that strain P39 formed a distinct cluster (Fig. 6), indicating that it represents a novel species within this genus. Based on these findings, we propose that P39 is a novel species belonging to the *Paraburkholderia* genus and was named *Paraburkholderia busanensis* sp. nov.

## Genomic analysis and annotation of *P. busanensis* P39 sp. nov.

The whole-genome sequence of *P. busanensis* sp. nov. P39 was assembled into two chromosomes consisting of 7,798,337 bp with a 63.81% G + C content. Accession

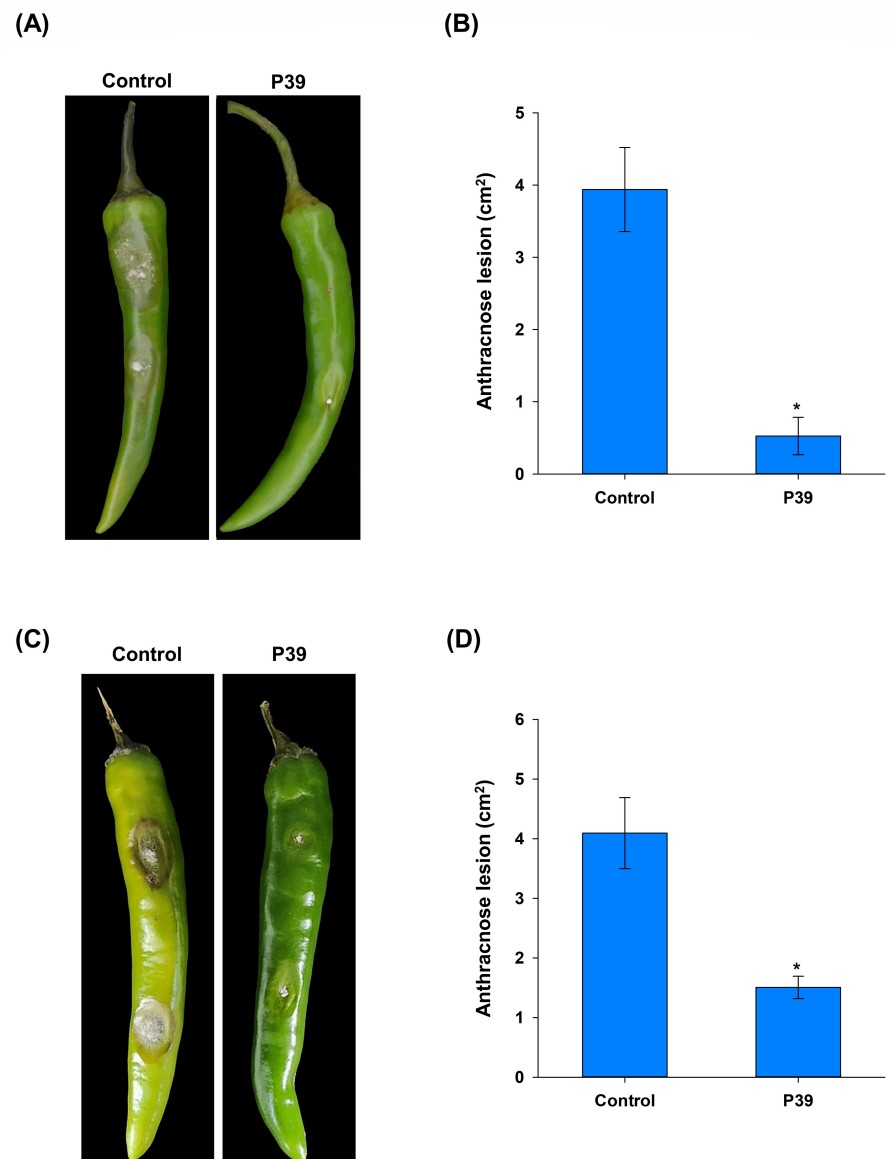

**FIG 4** *In vivo* assays on fresh green pepper for the effect of *Paraburkholderia* sp. nov. P39 and the produced volatiles of the development of anthracnose symptoms caused by *Colletotrichum scovillei* KC05. (A) Dipping assay in bacterial suspension of P39 for 2 h followed by inoculation with *C. scovillei* KC05. An amount of 10 mM $MgSO_4$ served as negative control. The photographs represent the development of the typical anthracnose symptoms on inoculated and treated fresh pepper 5 days after inoculation. (B) Bar graph of the evaluated anthracnose lesion area ($cm^2$) using ImageJ software on treated and inoculated pepper compared to the control. (C) The volatile assays on fresh green pepper inoculated with *C. scovillei* KC05 and incubated in a sealed square dish containing an open 90-mm petri dish smeared with P39 on nutrient agar (NA). Plates maintained with NA without bacterial smearing served as control. (D) Bar graph of the evaluated anthracnose lesion area ($cm^2$) using ImageJ software on treated and inoculated pepper compared to the control. The asterisks on the error bar represent significant differences according to a *t*-test at $P < 0.05$.

numbers CP058248 and CP058249 were assigned to contigs 1 and 2, respectively, for deposition in GenBank. Table 2 lists the basic features of the P39 genome, including the coding DNA sequences (CDSs), GC content, contigs, repeat regions, rRNA, hypothetical proteins, proteins with functional assignments, proteins with EC number assignments, proteins with GO assignments, and proteins with pathway assignments.

Based on the Rapid Annotation using Subsystem Technology (RAST) analysis of *P. busanensis* sp. nov. P39 genome, 7,246 coding DNA sequences were detected. Among them, 5,600 CDSs (77.2%) were not assigned to any specific subsystem, while 1,646 CDSs

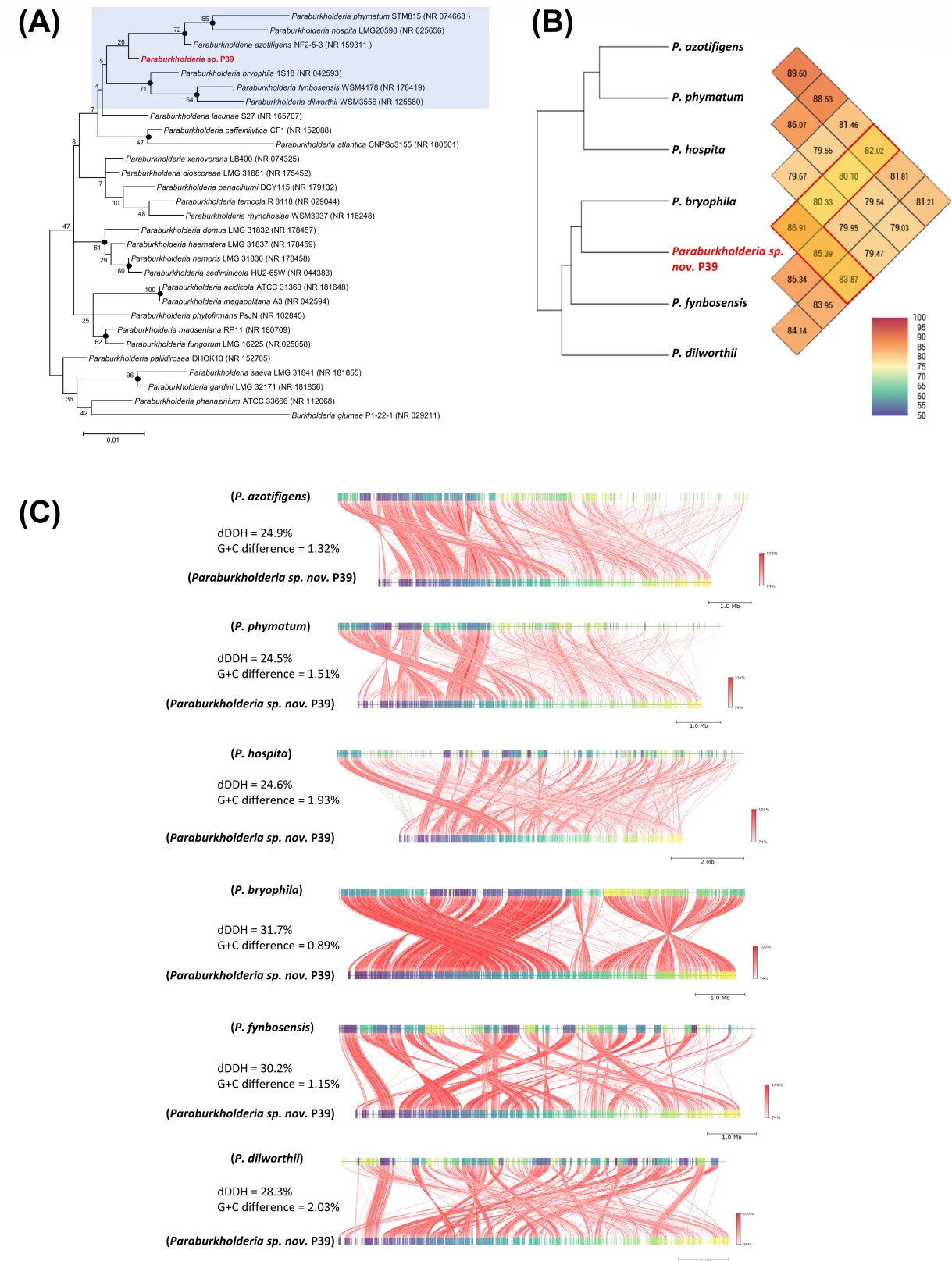

**FIG 5** Identification and *in silico* comparative genomics of *Paraburkholderia* sp. nov. P39 with related *Paraburkholderia* species. (A) Phylogenetic tree of the 16S rRNA gene sequence analysis, constructed by the maximum-likelihood method, showing the relationships between the selected *Paraburkholderia* sp. nov. P39 and the closely related type strains within the *Paraburkholderia* genus. The bootstrap values for 1,000 replications are shown at the branching points. The

**FIG 5** (Continued)

solid black dots on the branching points indicate that the corresponding nodes were also recovered in the trees constructed by the neighbor-joining method with bootstrap values (>50%). Numbers in parentheses indicate the accession numbers of the used type strains on the NCBI GenBank. The 16S rRNA gene sequence of *Burkholderia glumae* served as an outgroup. (B) Average nucleotide identity (ANI) pairwise comparison between the selected *Paraburkholderia* sp. nov. P39 and the six closely related species of *Paraburkholderia* that showed close similarity in the 16S rRNA analysis. (C) The genome alignment, digital DNA-DNA hybridization (dDDH), and G + C% difference between the selected *Paraburkholderia* sp. nov. P39 and the six closely related species of *Paraburkholderia*. Both the ANI and dDDH % values were less than the proposed and generally accepted species boundary for 95%–96% and 70%, respectively.

(22.8%) were categorized into various subsystems. The majority of the genes fell into the amino acid and derivative subsystems, accounting for 26.2% of the assigned genes, with 431 CDSs identified. The carbohydrate subsystem comprised 21.1%, with 348 CDSs assigned. Protein metabolism accounted for 13.8% of the assigned genes, represented by 227 CDSs. Cofactors, vitamins, prosthetic groups, and pigments encompassed 12.8% of the assigned genes with 210 CDSs. Membrane transport encompassed 3.67% of the assigned genes with 85 CDSs.

AntiSMASH analysis of *P. busanensis* sp. nov. P39 identified nine putative biosynthetic gene clusters associated with specialized metabolite production (Fig. 7). Four of the identified biosynthetic gene clusters showed low similarity to known clusters involved in the production of specific metabolites. The third cluster exhibited putative aryl polyene secondary metabolite production, sharing only 35% similarity with the aryl polyene gene cluster of *Aliivibrio fischeri* (20). Similarly, the fourth cluster was associated with putative non-ribosomal peptide synthetase-like secondary metabolite with a similarity level of 50% to the fragin gene cluster from *Burkholderia cenocepacia* (21). Moreover, the sixth cluster was associated with putative non-ribosomal peptide metallophores, non-ribosomal peptide synthetase, and terpene production, with a similarity level of 53% with the ornibactin gene cluster from *Burkholderia cepacia* (22). The eighth cluster exhibited putative phosphonate biosynthesis with a similarity level of only 6% to the oxazolomycin B gene cluster from *Streptomyces albus* (23).

In contrast, the remaining five putative secondary metabolite gene clusters associated with terpene, polyketide synthase-like, butyrolactone, hserlactone, and ribosomally synthesized and post-translationally modified peptide product-like linocin M18 showed no similarity with any known gene clusters. The low similarity levels of the detected putative gene clusters of secondary metabolites in *P. busanensis* sp. nov. P39 suggest the possibility of novel secondary metabolite production by this strain, warranting further investigation of its unique secondary metabolite profile. Putative secondary metabolite gene clusters detected in *P. busanensis* sp. nov. P39 are listed in Fig. 7.

## Chitin metabolism in *P. busanensis* sp. nov. P39: chitinase activity, related genes, and proposed mechanism

Confirmation of chitinase enzyme activity in the P39 strain was achieved through the conducted chitinase kit assay. The results of the assay revealed that the P39 strain exhibited endochitinase activity. The assessed endochitinase activity in the P39 culture extracts was not significantly different from the enzyme activity in broth supplemented

**TABLE 1** List of genomes from related species used for comparison in this study

| Species | Strain | BioSample | BioProject | Accession number | Base pairs | Percent G + C | No. of proteins | Reference |
|---|---|---|---|---|---|---|---|---|
| *Paraburkholderia* sp. | P39 | SAMN15207046 | PRJNA638731 | CP058248 CP058249 | 7,798,337 | 63.81 | 6,733 | This study |
| *P. phymatum* | LMG21445 | SAMN02598384 | PRJNA17409 | GCA_000020045.1 | 8,676,562 | 62.29 | 7,496 | (13, 14) |
| *P. fynbosensis* | LMG 27177 | SAMEA6678267 | PRJEB37806 | GCA_902859935.1 | 8,458,566 | 62.67 | 7,593 | (15) |
| *P. dilworthii* | WSM3556 | SAMN02440790 | PRJNA224116 | GCF_000472525.1 | 7,678,836 | 61.77 | 6,968 | (14, 16) |
| *P. hospita* | LMG20598 | SAMN07573571 | PRJNA224116 | GCF_002902965.1 | 11,198,650 | 61.88 | 10,428 | (14, 17) |
| *P. azotifigens* | NF 2-5-3 | SAMN12429752 | PRJNA558192 | GCA_007995085.1 | 9,704,342 | 62.48 | 8,254 | (18) |
| *P. bryophila* | LMG 23644 | SAMN28869333 | PRJNA846155 | GCA_028473685.1 | 8,007,187 | 62.9 | 7,095 | (14, 19) |

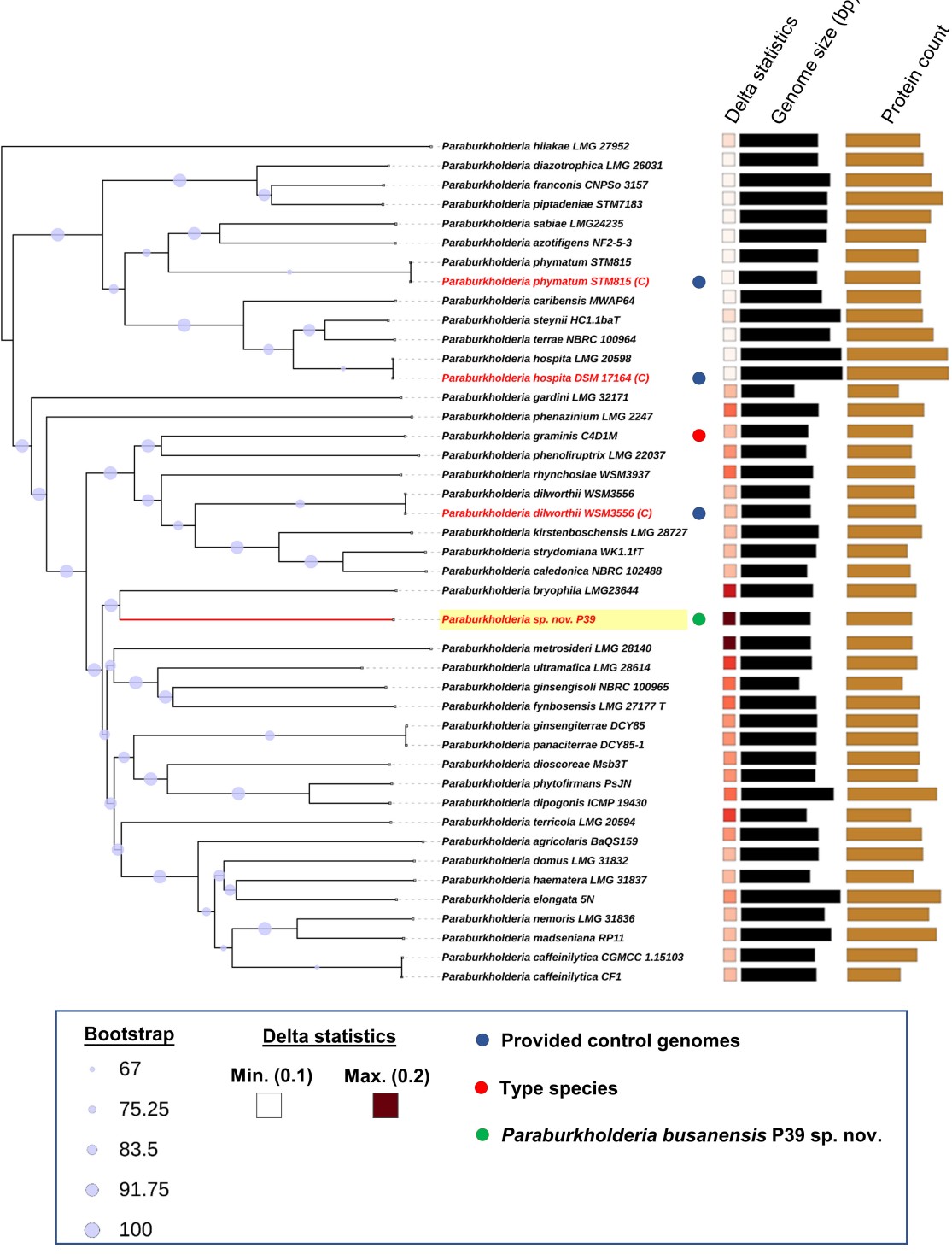

**FIG 6** Phylogenomic tree constructed on Type (Strain) Genome Server (https://tygs.dsmz.de/), inferred with FastME 2.1.6.1 , from genome GBDP distances calculated using whole-genome sequences of the selected *Paraburkholderia* sp. nov. P39 and the related species within *Paraburkholderia* genus. The branch lengths are scaled in terms of GBDP formula $d_5$. The pseudo-bootstrap values (>60%) are shown as purple nodes on the branching lines from 100 replications, with an average branch support of 99.8%. The tree was rooted at the midpoint. Three genomic sequences of *P. phymatum* LMG21445, *P. hospita* LMG20598, and *P. dilworthii* WSM3556 were provided as control.

with fungal mycelia (Fig. 8A). There was a slight decrease in endochitinase activity in media supplemented with fungal mycelia; however, this decrease was not statistically

**TABLE 2**   Genomic features of *Paraburkholderia busanensis* P39*a*

| Feature | Value |
| --- | --- |
| Length (bp) | 7,798,337 |
| GC % | 63.81 |
| Contigs | 2 |
| CDS | 7,243 |
| CDS ratio | 0.93 |
| Hypothetical CDS | 2,590 |
| Hypothetical CDS ratio | 0.443 |
| tRNA | 59 |
| Repeat regions | 50 |
| rRNA | 21 |
| Hypothetical proteins | 2,590 |
| Proteins with functional assignments | 4,653 |
| Proteins with EC number assignments | 1,395 |
| Proteins with GO assignments | 1,211 |
| Proteins with pathway assignments | 1,082 |

*a*Values were generated following genome analysis and annotation using the Bacterial and Viral Bioinformatics Resource Center (https://www.bv-brc.org/).

significant. It is plausible that this decline in activity is associated with the utilization of chitin present in the fungal cell wall.

Using genome mining to detect the genes involved in chitin metabolism and utilization, eight genes were identified, including five adjacent genes that formed a cluster. The other three genes are located elsewhere in the genome and include genes

**FIG 7**   Putative secondary metabolites in *Paraburkholderia busanensis* P39 predicted using antiSMASH (version 7). PKS-like, polyketide synthase-like; NRPS-like, non-ribosomal peptide synthetase-like; NRPS, non-ribosomal peptide synthetase; NRP-metallophore, non-ribosomal peptide metallophore; RiPP-like, ribosomally synthesized and post-translationally modified peptide product. Gene cluster keys: gray, other genes; red, core biosynthetic genes; orange, additional genes; purple, regulatory genes; green, transport-related genes.

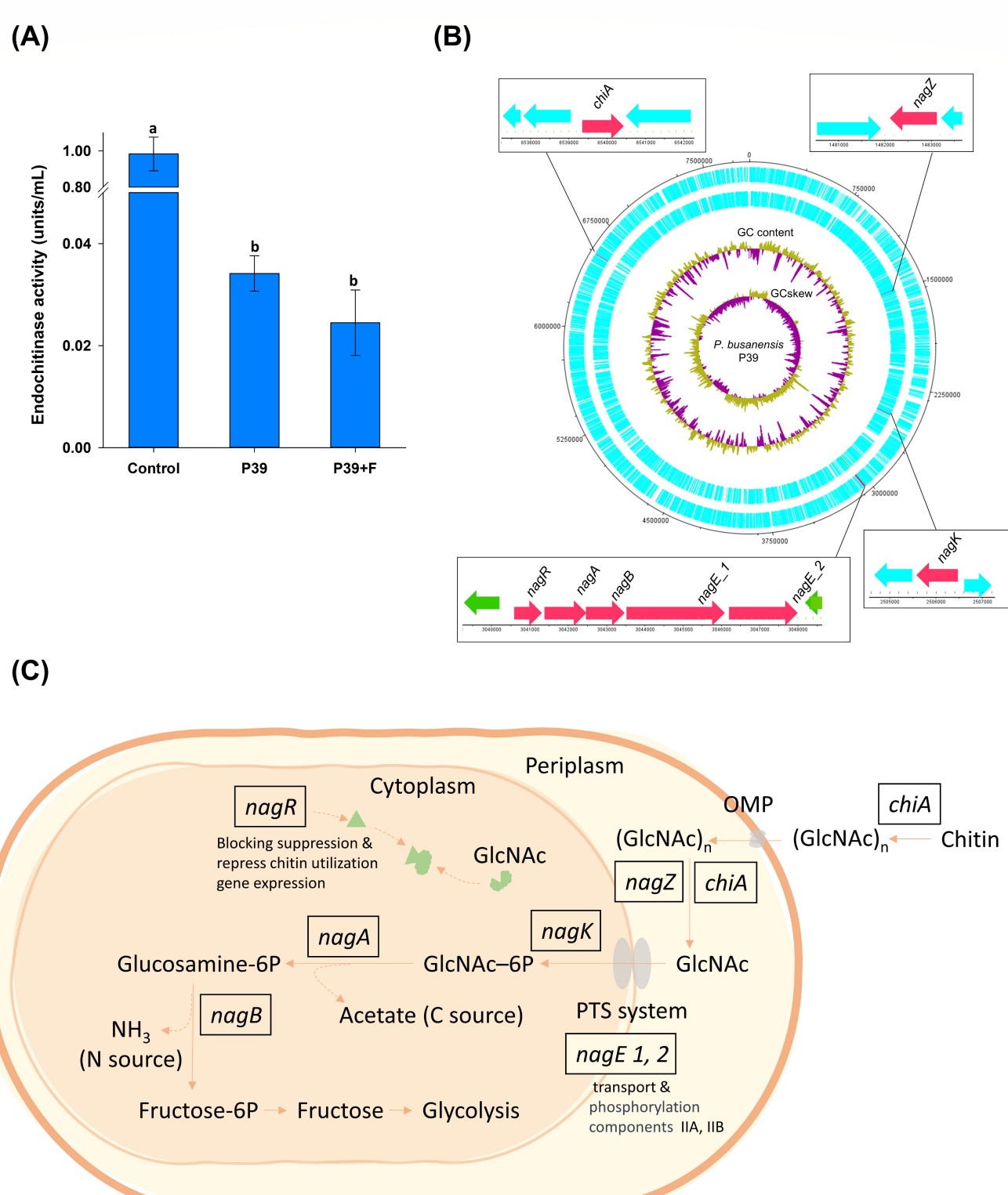

**FIG 8** Chitin utilization in *Paraburkholderia busanensis* P39 and proposed mechanism of utilization. (A) Endo-chitinolytic activity of *P. busanensis* P39 in nutrient broth (NB), compared to NB supplemented with fungal mycelial of *Colletotrichum scovillei* and the chitinase enzyme control. The bars represent the mean values and standard error of three replicates. Different letters on the error bars indicate significant differences according to the least significant difference test at a significance level of $P < 0.05$. (B) The genes related to chitin utilization and their position on the circular map of the *P. busanensis* P39 genome. (C) The proposed

**FIG 8** (Continued)

chitin utilization process in *P. busanensis* P39, beginning with the *chiA* gene, which produces the chitinase enzyme that breaks down chitin into monomers and less complex polymers of N-acetylglucosamine $(GlcNAc)_n$. The GlcNAc polymers and monomers are transported through the outer membrane proteins (OMPs) to the periplasmic space, where the *nagZ* and *chiA* genes further break down the polymers into monomers. Subsequently, the GlcNAc monomers are transported into the cytoplasm through the phosphotransferase system (PTS) with the help of the *nagE1* and *nagE2* genes, which are responsible for the transport and phosphorylation components IIA and IIB. The *nagK* gene catalyzes the phosphorylation of GlcNAc into N- (GlcNAc–6P), and the resulting is transformed to glucosamine 6-phosphate by the *nagA* gene. This process releases acetate, which can be used by the bacterial cell as a carbon source. The glucosamine 6-phosphate is then transformed to fructose 6-phosphate by the *nagB* gene. This process releases $NH_3$, which serves as a nitrogen source for the bacterial cell. The fructose produced then enters glycolysis. The *nagR* regulator gene encodes the NagR protein that regulates the chitin utilization process. The presence of GlcNAc inhibits NagR repression of the chitin utilization gene expression, allowing the process to proceed efficiently in the presence of the substrate, as in the absence of GlcNAc, NagR binds to its target DNA sequences and inhibits the expression of GlcNAc utilization genes, conserving cellular resources and energy.

responsible for chitinase synthesis (*chiA*), beta-N-acetylglucosaminidase synthesis (*nagZ*), and N-acetylglucosamine kinase synthesis (*nagK*) (Fig. 8B).

The presence of chitin metabolism and utilization-related genes within the P39 genome offers compelling evidence for a potential pathway involved in chitin utilization, as illustrated in Fig. 8B. Drawing upon the detected genes and existing research on bacterial amino-sugar metabolism (24–26), we have put forth a proposed pathway. The process begins with chitinase, which is produced by the *chiA* gene, breaking down chitin into less complex polymers of N-acetylglucosamine (GlcNAc) and GlcNAc monomers that enter the bacterial cell via outer membrane proteins. The GlcNAc polymers are further broken down into monomers in the periplasm by the N-acetylglucosamine kinase produced by the *nagZ* gene and chitinase. GlcNAc monomers are then transferred into the cytoplasm by the phosphotransferase system and the transport and phosphorylation components IIA and IIB, which are encoded by *nagE1* and *nagE2*. N-acetylglucosamine kinase, encoded by *nagK*, catalyzes the phosphorylation of GlcNAc into GlcNAc-6P.

In the cytoplasm, N-acetylglucosamine-6-phosphate deacetylase encoded by *nagA* converts GlcNAc-6-P to glucosamine-6P, thereby releasing acetate, which serves as a carbon source for the bacterium. The glucosamine-6P is converted into fructose-6-phosphate by the glucosamine-6-phosphate deaminase encoded by *nagB*, releasing $NH_3$ that can be used by the bacterium as a nitrogen source. Fructose then enters glycolysis, enabling the bacterium to generate ATP, which is the primary energy source for many cellular processes, as well as precursor molecules that can be used for the biosynthesis of other essential molecules required for survival and growth.

Chitin metabolism is regulated by the NagR protein encoded by the *nagR* gene, which generally binds to target DNA sequences and inhibits the expression of GlcNAc utilization genes in the absence of GlcNAc. GlcNAc forms a complex with the NagR protein, inhibiting the repression of gene expression and allowing the process to proceed effectively. This may be to conserve cellular resources and energy, as the process works only in the presence of a substrate.

## Evidence on mycophagy of *P. busanensis* sp. nov. P39 against *C. scovillei* KC05

Based on the nutrient broth (NB) mycophagy assay, *P. busanensis* sp. nov. P39 showed a slight change in the presence of fungal mycelia. Therefore, NB was diluted to 10×, 100×, and 1,000×, and the bacterial population showed a significant increase at 100× and 1,000 × dilutions as shown in the photographs of the bacterial spots at $10^{-3}$ to $10^{-6}$ dilutions and as estimated by counting represented at CFU per milliliter (Fig. 9A and B). It could be inferred that when the broth was more diluted with less available nutrients for bacterial growth, the bacteria might have utilized the supplemented fungal mycelia for growth and increased their populations. In the same context, the fungal mycelial dry weight assessed following incubation showed a significant reduction, and this reduction was even more significant at 1,000×, which is in accordance with the assumption that the greater the dilution of the broth, the more fungal mycelia are utilized as nutrient sources (Fig. 9C).

**(A)**

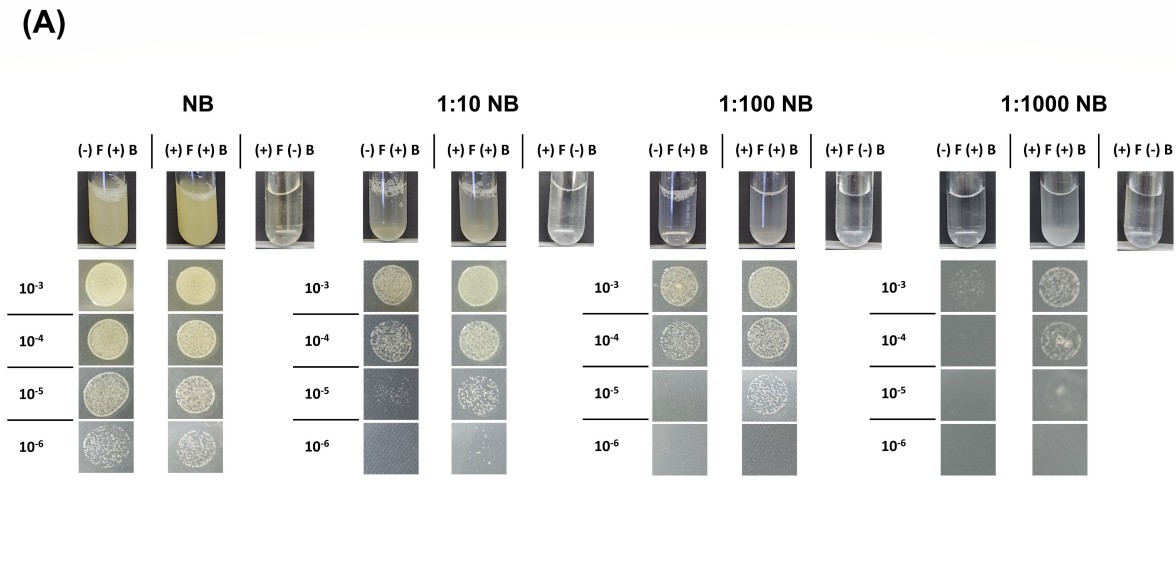

**(B)**

**(C)**

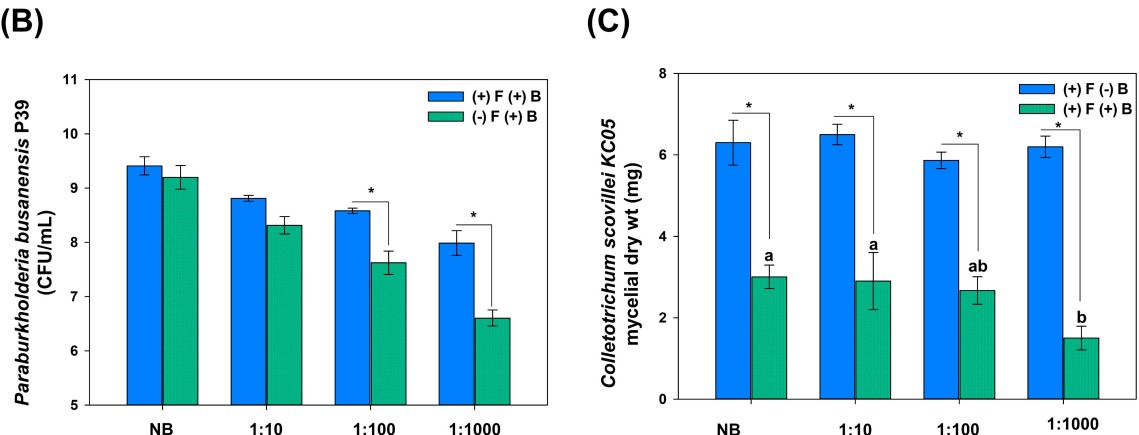

**FIG 9** Mycophagy assay of *Paraburkholderia busanensis* P39 against *Colletotrichum scovillei* KC05 in normal and diluted nutrient broth. (A) Photographs of the culture tubes in NB and diluted NB (1:10, 1:100, and 1:100 dilutions) containing 100 µL from bacterial suspension (OD$_{600}$ = 0.6) and 100 µL from washed fungal mycelia (6 mg/mL). One day after incubation with shaking at 30°C, 5 µL from $10^{-3}$ to $10^{-6}$ serial dilutions were spotted on nutrient agar and photographed after 2 days of incubation. (B) Bar graphs represent the assessed bacterial population from the mycophagy assay as CFU per milliliter between fungal-treated and untreated tubes [(+)F and (−)F, respectively]. Data presented are the means and standard errors of three replicates, and the asterisk on the error bar indicates a significant difference according to a *t*-test at *P* < 0.05 between fungal-treated and untreated tubes at each broth dilution. (C) The evaluated mycelial dry weight of *C. scovillei* KC05 from the mycophagy assay represented as mycelial dry weight (mg). Data presented are the means and standard errors of three replicates, and the asterisk on the error bar indicates a significant difference according to a *t*-test at *P* < 0.05 between bacteria-treated and untreated tubes [(+)B and (−)B, respectively] at each broth dilution. The different lowercase letters represent significant differences between the mycelial dry weights of *C. scovillei* KC05 in bacteria-treated samples at NB and diluted NB tubes according to the least significant difference test at *P* < 0.05.

Microscopic observations were made in tubes containing *C. scovillei* KC05 mycelia in the presence of *P. busanensis* sp. nov. P39 and compared with *C. scovillei* KC05 alone in diluted broth 10×, 100×, and 1,000×. In the absence of bacteria, the fungal mycelia remained intact, and their cellular constituents were visible within the fungal hyphae (Fig. 10A). However, in the presence of *P. busanensis* sp. nov. P39, the fungal mycelia appeared damaged and depleted of cellular contents at multiple points of rupture, indicating the action of the bacteria in breaching the fungal cell wall and possibly feeding on the cellular contents (Fig. 10B). This observation for *P. busanensis* sp. nov. P39, which causes extensive damage to *C. scovillei* KC05 mycelia, could mechanistically explain the previously observed reduction in fungal mycelial dry weight and increase in the bacterial population, especially in diluted broth. In addition, we observed the

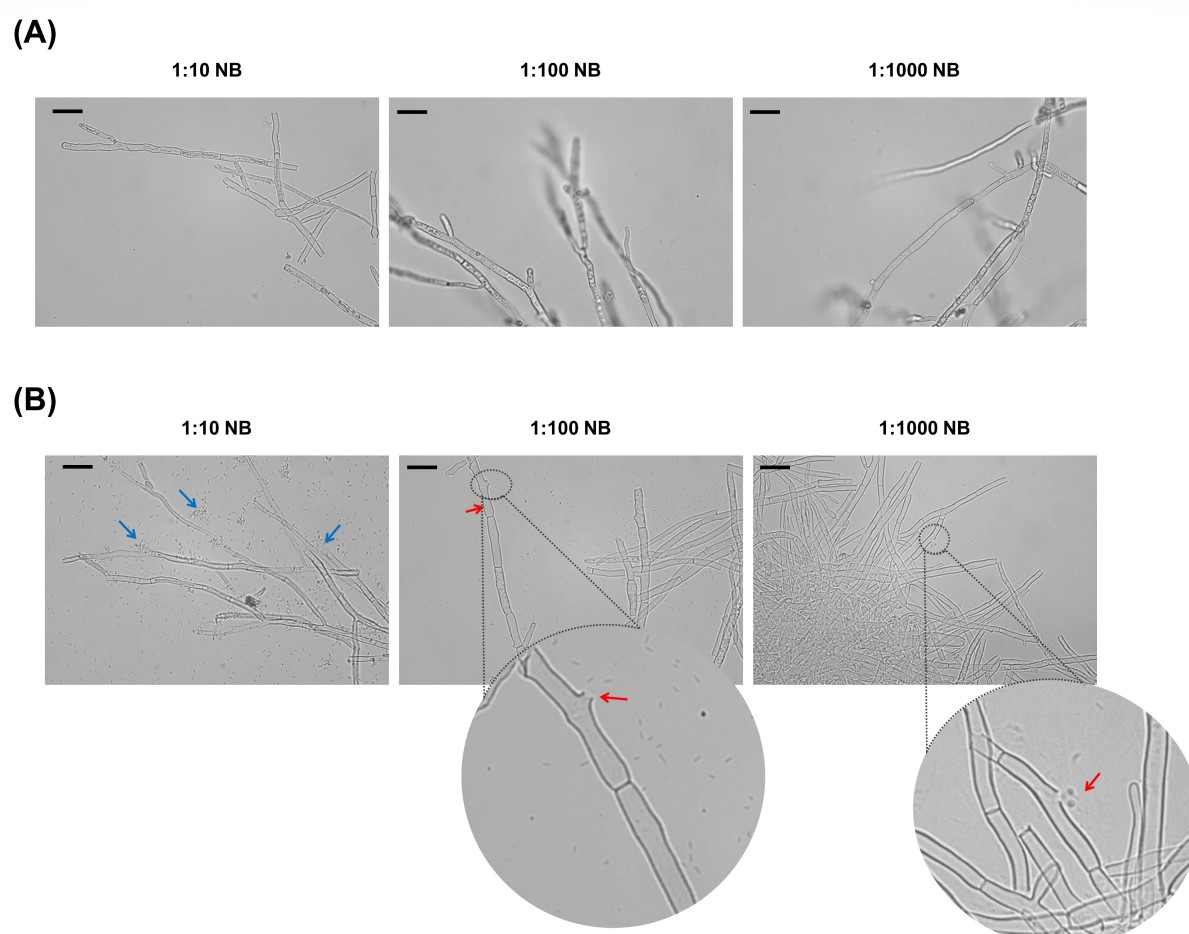

**FIG 10** Light microscope images of *Paraburkholderia busanensis* P39 against *Colletotrichum scovillei* KC05 in nutrient broth. (A) Control fungal mycelia without treatment with *P. busanensis* P39, showing intact mycelia with undepleted cellular contents and without cell wall damage. (B) Fungal mycelia in the presence of *P. busanensis* P39, demonstrating damaged mycelia that are not intact and depletion of cellular contents at multiple points. Enlarged circular images depict points of rupture in the mycelia, indicating the action of the bacteria in breaching the fungal cell wall and possibly feeding on cellular contents. The red arrows indicate points of fungal cell wall rupture, and the blue arrows highlight the accumulation of concentrated bacterial cells around damaged fungal mycelia. The scale bar indicates 20 µm.

clustering or grouping of *P. busanensis* sp. nov. P39 around the fungal hyphae, especially at higher concentrations, indicating a possible attraction to the fungal mycelia (Fig. 10B).

## Characteristics of *P. busanensis* sp. nov. P39

In the *in vitro* assays evaluating the activities of the P39 strain, it was observed that strain P39 exhibited a weak ability to degrade cellulose. Furthermore, the strain showed positive results for phosphate solubilization, protease production, and siderophore production (Fig. S1).

The phenotypic and chemotaxonomic characteristics of strain P39 were consistent with those of *Paraburkholderia*. Colonies on nutrient agar (NA) after 2 days of incubation at 28°C were creamy or pale yellow, circular, smooth, entire, convex, and 1.0–1.5 mm in diameter. The cells were motile and Gram negative. Optimum growth occurred at 28°C, and no growth was observed at 10°C and 40°C. At optimum temperature, growth was observed in the presence of 0%–5% (wt/vol) NaCl and at pH 5.0–8.0. The differential characteristics of strain P39 compared to closely related species of *Paraburkholderia* are shown in Table 3.

**TABLE 3** Differential characteristics of the isolated *Paraburkholderia* sp. nov. P39 strain in comparison to closely related species of *Paraburkholderia*[a]

| Characteristic | 1 | 2 | 3 | 4 | 5 | 6 | 7 |
|---|---|---|---|---|---|---|---|
| Isolation source | Pine soil | Paddy soil | Soil | Root nodule | Root nodule | Root nodule | ND |
| Colony color | Pale yellow | Cream | White | Milky | White | White | ND |
| Optimum temperature (°C) | 28 | 15–37 | 28 | 28 | 15–37 | ND | ND |
| pH range | 5–8 | 6–8 | ND | ND | ND | 5.5–8 | ND |
| NaCl range (%, wt/vol) | 0–5 | 0–2 | 0–0.5 | ND | 0–10% | ND | ND |
| Oxidase activity | + | + | + | + | ND | ND | + |
| Reduction of nitrate to nitrite | + | + | + | + | + | – | – |
| Arginine dihydrolase | – | ND | ND | – | W | + | – |
| Tryptophan deaminase | – | ND | ND | – | + | W | ND |
| Urease | – | | | – | – | – | |
| β-Galactosidase | + | + | + | + | W | + | – |
| β-Glucosidase | + | + | + | – | ND | ND | |
| Assimilation of: | | | | | | | |
| Glucose | + | W | + | + | + | + | + |
| Capric acid | + | + | – | + | – | + | + |
| Mannitol | + | ND | ND | + | + | + | + |
| Mannose | + | ND | ND | + | + | + | + |
| Arabinose | + | ND | ND | + | + | + | + |
| N-acetylglucosamine | + | ND | ND | + | + | + | + |
| Adipic acid | – | ND | ND | – | + | + | |
| Enzyme activity: | | | | | | | |
| Esterase lipase (C8) | + | + | – | + | ND | ND | ND |
| Valine arylamidase | + | + | + | + | ND | ND | ND |
| Cystine arylamidase | + | + | + | + | ND | ND | ND |
| DNA G + C content (mol%) | 63.8 | 64.2 | 62.0[c] | 62.1[e] | 61.4–61.9 | 62.4 | 62 |

[a]Strains: 1, *Paraburkholderia* sp. nov. P39 (this study); 2, *P. azotifigens* NF2-5-3[T] (18); 3, *P. hospita* LMG20598[T] (17); 4, *P. phymatum* LMG21445[T] (13); 5, *P. dilworthii* WSM3556[T] (16); 6, *P. fynbosensis* WSM4178[T] (16); 7, *P. bryophila* LMG 23644[T] (19, 27). +, positive; W, weakly positive; –, negative; ND, no available data.

The predominant fatty acids were summed feature 2 (iso-C16:1 I/C14:0 3-OH), C17: 0cyclo, C19:0 cyclo ω8c, C16:0, and summed feature 8 (C18:1 ω7c/C18:1 ω6c). The complete cellular fatty acid composition is given in Table 4 in comparison with the

**TABLE 4** Cellular fatty acids composition of the isolated *Paraburkholderia* sp. nov. P39 in comparison to closely related *Paraburkholderia* spp.[a]

| Fatty acid | 1 | 2 | 3 | 4 | 5 | 6 | 7 |
|---|---|---|---|---|---|---|---|
| $C_{12:0}$ | 1.61 | 1.1 | 1.3 | tr | – | – | – |
| $C_{14:0}$ | 2.61 | 5.3 | 2.9 | 4.2 | 4.3 | 4.3 | 1 |
| $C_{16:0}$ | 11.8 | 25.8 | 25 | 30.1 | 20.3 | 17.8 | 22.6 |
| $C_{18:0}$ | 0.51 | 1.8 | 1.3 | 1.6 | tr | tr | 1.5 |
| $C_{16:0}$ 2-OH | 0.93 | 2.2 | 2.2 | 1.2 | 1.3 | 1.4 | 3.9 |
| $C_{16:0}$ 3-OH | 4.06 | 5.4 | 4.6 | 4.7 | 5.9 | 5.6 | – |
| $C_{16:1}$ 2-OH | 0.93 | 1.6 | 1.2 | 1 | 2.3 | 2.6 | 2.1 |
| $C_{18:1}$ 2-OH | 2.87 | – | – | 0.7 | tr | nd | – |
| $C_{17:0}$ cyclo | 12.95 | 10.5 | 20.4 | 21.1 | 9.3 | 11.1 | 11 |
| $C_{19:0}$ cyclo ω8c | 12.9 | 6.7 | 11.8 | 13.2 | 3.4 | 2.5 | 2.7 |
| SF[b] 2; iso-$C_{16:1}$ I/$C_{14:0}$ 3-OH | 20.95 | 7.9 | 4.6 | 5.3 | 6.7 | 7.3 | – |
| SF 3; $C_{16:1}$ ω7c/$C_{16:1}$ω6c | 3.46 | 6.2 | 3.6 | 3.1 | 17.5 | 17 | 22.2 |
| SF 8; $C_{18:1}$ ω7c/$C_{18:1}$ω6c | 10.15 | 22.9 | 20.4 | 13.5 | – | – | 22.6 |
| SF 9; $C_{18:1}$ ω7c 11-methyl | 0.24 | 1.2 | – | – | – | – | – |

[a]Strains: 1, *Paraburkholderia* sp. nov. P39 (this study); 2, *P. azotifigens* NF2-5-3[T] (18); 3, *P. hospita* LMG20598[T] (18); 4, *P. phymatum* LMG21445[T] (18); 5, *P. dilworthii* WSM3556[T] (16); 6, *P. fynbosensis* WSM4178[T] (16); 7, *P. bryophila* LMG 23644[T] (27).
[b]Summed features: groups of two or three fatty acids that could not be separated by Gas-Liquid Chromatography (GLC) using the Microbial IDentification (MIDI) system.

phylogenetically closest *Paraburkholderia* spp. The major respiratory quinone in the P39 strain was ubiquinone 8 (Q-8), similar to that in members of the *Paraburkholderia* genus. The polar lipids identified were phosphatidylglycerol, phosphatidylethanolamine, unidentified amino lipids, and unidentified polar lipids.

## Description of *Paraburkholderia busanensis* sp. nov.

*Paraburkholderia busanensis* (bu.san.en′sis. N. L. adj. *busanensis*) was obtained from Busan, Korea, where the type strain was isolated). The cells are Gram negative, rod shaped, motile, and measure approximately 0.8 × 2.9 µm. After 2 days of incubation at 28℃, colonies on NA displayed a creamy or pale-yellow color, circular shape, smooth texture, entire edges, convex profile, and diameters ranging from 1.0 to 1.8 mm. The organism exhibited a growth range of 15–37℃, with an optimum temperature of 28℃, and a pH range of 5.0–8.0, with an optimum of 7.0. Furthermore, this bacterium can grow without the addition of NaCl and in the presence of up to 5% (wt/vol) NaCl on R2A agar.

The bacterium is catalase- and oxidase positive, can hydrolyze casein, solubilize insoluble phosphate, and displays a weak ability to degrade CM cellulose. Endochitinase activity was confirmed in strain P39, while no exochitinase activity was detected. ANI 20NE and API ZYM tests confirmed the ability of the bacteria to reduce nitrate to nitrite, -galactosidase, -glucosidase, hydrolyze gelatin, and assimilate glucose, capric acid, mannitol, mannose, arabinose, N-acetylglucosamine, esterase (C4), esterase lipase (C8), lipase (C14), alkaline phosphatase, valine arylamidase, and cystine arylamidase. The bacteria tested negative for arginine dihydrolase, tryptophan deaminase, urease, and adipocytic acid.

The predominant respiratory quinone of the bacterium is ubiquinone Q8, while the major fatty acids include summed feature 2 (iso-C16:1 I/C14:0 3-OH), C17:0cyclo, C19:0 cyclo ω8c, C16:0, and summed feature 8 (C18:1 ω7c/C18:1 ω6c). The polar lipid profile of an organism is composed of phosphatidylglycerol, phosphatidylethanolamine, unidentified amino lipids, and unidentified polar lipids. Finally, strain P39 was deposited in Korea Culture Center of Microorganisms (KCCM; KFCC11965P). The G + C content of the deposited strain was 63.8%.

## DISCUSSION

In the present study, a novel approach was adopted to identify highly effective bacterial strains with biocontrol properties against *C. scovillei* causing pepper anthracnose. Traditionally, the process of selecting and identifying biocontrol strains involves the isolation of a large number of microbes followed by *in vitro* screening. Accordingly, taxonomic information on the selected strains is then collected (28). Although this approach has proven successful in selecting efficient biocontrol strains and can help avoid unnecessary taxonomic work, it has limitations. One of the limitations is the potential selection of a biocontrol bacterium that exhibits activity against plant pathogens but also belongs to a group of bacteria known to be pathogenic, toxic, or harmful to the environment. The soil and rhizosphere, which are significant sources of biocontrol agents, are also considered reservoirs of opportunistic human pathogens (29).

The classification of a bacterium as taxonomically related to pathogenic or opportunistic pathogens raises concerns because there are restrictions on the registration of such strains as plant growth-promoting or biocontrol agents. It is important to note that the isolated strain may not be pathogenic or opportunistic; however, its interaction with related species could potentially transfer fitness features to those that are pathogenic or opportunistic. A notable example of this concern is the risk assessment conducted by the United States Environmental Protection Agency for the biopesticide *Burkholderia cepacia*, a strain belonging to a species associated with cystic fibrosis in humans. This highlights the need for careful evaluation and regulatory scrutiny when considering the use of taxonomically related strains with potential biocontrol activities (30).

To address these concerns, we propose an alternative approach that incorporates taxonomically guided selection. By focusing on a specific taxonomic group of bacteria

known to harbor potential biocontrol strains with beneficial properties, we can narrow our search and minimize the risk of selecting harmful organisms, thereby avoiding the need for laborious *in vitro* screening assays. This is particularly important for our research, which focuses on postharvest treatment because the application of a safe bioagent is crucial when dealing with food (7). In the present study, we specifically targeted the *Paraburkholderia* group, which is renowned for its beneficial effects on plants and possesses large genomes that enable versatility and adaptation to diverse environments (31). This selection ensured that we harnessed the potential of bacteria with a proven track record of positive effects, increasing the likelihood of successful biocontrol. The genus *Paraburkholderia* was proposed as a distinct lineage separated from the others within the *Burkholderia sensu lato* group. It includes members that exhibit potential beneficial plant traits and other advantageous characteristics. Additionally, the absence of virulence loci, which are typical markers of pathogenicity in mammalian pathogens, was evidenced by genomic analyses and functional pathogenicity tests conducted on beneficial plant members (14, 32, 33).

To conduct the taxonomy-guided screening, we utilized partial 16S rRNA sequencing, which provided the information required to assign the isolated bacteria at least to the genus level. The type strain of *Paraburkholderia* served as a reference to guide the selection process. Consequently, 33 bacterial strains taxonomically related to *Paraburkholderia* were selected and further analyzed to identify the most efficient biocontrol agent. Among these strains, P39 exhibited remarkable control activity against *C. scovillei*. Therefore, this strain was characterized as the most promising candidate for further investigation and application in biocontrol strategies.

Analysis of the 16S rRNA sequence was unsuccessful to assign strain P39 to a species within the *Paraburkholderia* genus. Consequently, the whole-genome sequence was obtained to ensure accurate identification and to gain a deeper understanding of the observed biocontrol activities. Genomic *in silico* comparative analysis and examination of genome-to-genome relatedness with closely related type strains, along with biochemical and physiological characterization, revealed that the selected strain was a novel species within the *Paraburkholderia* genus, and the name *P. busanensis* was assigned.

The selected strain exhibited multiple forms of bioactivity against *C. scovillei*, as demonstrated by *in vitro* and *in vivo* assays involving pepper fruits. Specifically, the selected strain produced antifungal compounds that diffused through the agar and effectively inhibited fungal growth. The antagonistic activity and production of antifungal metabolites were further confirmed in a liquid culture containing fungal conidia and cells, or cell-free culture extracts of the selected strain. These findings were further supported by dipping assays conducted on inoculated pepper fruits. Previous studies have explored the use of bacterial biocontrol agents for postharvest diseases in various products. An early study by Korsten et al. (34) reported the antagonistic activity of bacterial strains from *Bacillus* and *Pseudomonas* spp. against various postharvest fungal pathogens and their efficacy in controlling anthracnose in inoculated avocado fruits. More recently, a strain of *Streptomyces* was isolated and found to show strong antifungal activity and produce various antifungal compounds against *Colletotrichum fragariae*, which causes postharvest decay of strawberries, as treatment of the fruits resulted in a reduced decay rate of strawberries and maintained fruit quality (35). Another strain of *Streptomyces* was found to show efficient control comparable to that of the chemical fungicide Benlate against *Colletotrichum siamense* causing anthracnose on mango fruits (36). A strain of *Burkholderia* was also reported to have antagonistic activity, reducing mycelial growth and conidial germination against *Colletotrichum orbiculare*, which causes anthracnose in cucumbers and has been suggested as a promising biocontrol agent (37).

Furthermore, the volatiles released by strain P39 lead to suppressed *in vitro* fungal growth and conidia production, which are responsible for the spread of infection on the affected fruits and reduce the development of anthracnose symptoms in inoculated pepper fruits *in vivo*. The production of antifungal volatiles, which are small, diffusible

organic compounds with low molecular weights, is considered important, especially for reporting biocontrol agents against postharvest diseases. This represents a potentially applicable fumigation treatment for preventing anthracnose development in harvested fruits. Previous studies have reported similar activity of bacterial biocontrol agents against *Colletotrichum* spp., as the volatiles produced by *Streptomyces philanthi* were effective in reducing fungal growth and disease development in pepper fruits (38).

Genomic analysis revealed features supporting the efficiency of the selected strain in terms of fitness, versatility, and biocontrol activity. The selected strains had a relatively large genome with a large portion of coding regions and many annotated features and functions, such as amino acids and derivatives, carbohydrates, protein metabolism, fatty acids, lipids, isoprenoids, membrane transport, including various types of secretion systems, and motility and chemotaxis. A notable proportion of the subsystem was allocated to membrane transport, encompassing various clusters of secretion systems, notably type II, type III, and type VI secretion systems. Detailed studies are essential to elucidate the functionality of these secretion systems in the P39 bacteria. Furthermore, understanding their potential role in biocontrol activities and interactions with fungi will be of significant interest. AntiSMASH analysis revealed that the selected strain exhibited the potential for the identification of novel secondary metabolites, as several metabolites with no similarities to known compounds were found. Among these, four clusters that exhibited low similarity to known clusters involved in the production of specific secondary metabolites were identified. These clusters include putative aryl polyenes, fragin, ornibactin, and oxazolomycin B clusters (20–23).

Aryl polyenes are yellow pigments produced by bacteria that act as fitness factors to protect against oxidative stress and contribute to biofilm formation (39, 40). Fragin, primarily produced by *Burkholderia cenocepacia*, is an antifungal compound with strong activity against fungi. It has been suggested that fragin functions as a metallophore, with metal chelation potentially being the molecular basis for its antifungal activity (21). Ornibactin, an important siderophore produced by members of *Burkholderia*, contributes significantly to the antifungal activity of microbes. By facilitating the scavenging of iron from the environment, ornibactin makes iron available for producing cells while inhibiting the growth of other microbes through iron depletion (41, 42). Oxazolomycin B was first reported in 1998 as a product of *Streptomyces albus* along with other compounds belonging to the oxazolomycin family of antibiotics that have been associated with various bioactivities (43, 44).

The low similarity to known clusters observed in these putative secondary metabolite gene clusters, along with the presence of several secondary metabolite gene clusters that exhibited no similarity to any known clusters, indicates that the isolated novel species *P. busanensis* P39 is a promising candidate for the characterization and isolation of novel secondary metabolites. These metabolites are likely to play important biological roles such as the strong antifungal potential reported in the current study.

The findings of the mycophagy assay demonstrate that bacteria can consume fungal mycelial masses, leading to population growth as they derive nutrients from fungi. Microscopic observations revealed damage and rupture at multiple locations in the fungal mycelia in the presence of the selected strain, resulting in the depletion of cellular contents within the fungal hyphae. Similar mycophagy has been previously reported in other bacterial species. The ability of certain bacteria, such as *Bacillus* species (e.g., *B. cereus* and *B. megaterium*), and *Pseudomonas* sp. to utilize fungal mycelia of *Fusarium* as a sole carbon source was first reported in 1961. These bacteria are referred to as mycophagous bacteria, actively accessing fungal nutrients primarily by causing leakage of fungal membranes (45). Several other studies have also reported similar findings in different bacterial species, employing various mechanisms including extracellular necrotrophy, extracellular biotrophy, and endocellular biotrophy (46). According to our results and microscopic observations, we propose that the selected strain *P. busanensis* P39 sp. nov. employs an extracellular necrotrophic strategy, utilizing lytic enzymes such as endochitinase to degrade the fungal cell wall and feed on the efflux of hyphal content.

This hypothesis is supported by the visible spots of rupture and depletion of cellular contents observed during our experiments.

In previous research, bacteria belonging to the genus *Collimonas*, such as *C. fungivorans*, have been reported to exhibit mycophagy activity. They can utilize fungal hyphae as a sole carbon source, apparently using fungal exudates as a signal molecule to locate hyphae (47–49). Methods for baiting and enriching mycophagous bacteria from the rhizosphere have also been proposed (50). In this study, bacterial isolates closely related to *Pseudomonas protegens* and *Burkholderia sp*. exhibiting mycophagy ability were isolated, suggesting that bacterial mycophagy is an important growth strategy among soil microbes. More recently, rice-associated *Burkholderia gladioli* was found to exhibit mycophagy. This ability was attributed to the utilization of a type III secretion system, which facilitated the translocation of a prophage tail-like protein effector into fungal mycelia, thereby demonstrating broad-spectrum antifungal activity (51).

Furthermore, the annotated genome of strain P39 was explored to investigate the potential genes involved in chitin and N-acetylglucosamine metabolism and consumption, which are prominent components of fungal cell walls. This analysis revealed the presence of a gene cluster dedicated to the utilization of N-acetylglucosamine, as well as the *chiA* gene responsible for chitinase production and transport-related genes. The endochitinase activity of the selected strain, P39, was also experimentally confirmed.

Based on the detection of these genes within the P39 genome, in conjunction with the available literature on chitin utilization processes, we proposed a putative pathway for the utilization of chitin and N-acetylglucosamine, supported by an *in vitro* mycophagy assay. This pathway helps explain, at least in part, the observed reduction in fungal mycelial weight accompanied by an increase in the bacterial population, particularly in nutrient-poor media. Other lytic enzymes could also take part in the process of utilizing fungal biomass and rupturing the fungal cell walls, including proteases and cellulases, which were found to be positive in the *in vitro* assay for the selected strain in this study. Microscopic observations further substantiated these findings, demonstrating the rupture, damage, and depletion of the fungal cell wall constituents.

Taken together, this study presents a novel taxonomy-guided approach for selecting effective biocontrol bacterial strains, streamlining the search for beneficial candidates, and avoiding laborious *in vitro* assays that may inadvertently lead to the selection of strains with affinities toward harmful or opportunistic pathogens. Using this approach, a bacterial strain exhibiting strong biocontrol activity against *C. scovillei*, the causal agent of pepper anthracnose, was identified. Genomic, biochemical, and physiological characterizations confirmed that the strain was a novel species within the *Paraburkholderia* genus. Genomic analysis revealed potentially beneficial traits and highlighted the potential for secondary metabolite production. This strain displays a unique interaction with fungi, consuming and utilizing fungal mycelia, possibly facilitated by genes related to chitin and N-acetylglucosamine metabolism. This novel species, *P. busanensis* P39, shows promise as an efficient biocontrol agent against pepper anthracnose and warrants further investigation into bacteria-fungi interactions and novel bioactive secondary metabolites.

## MATERIALS AND METHODS

### Bacterial isolation and taxonomy-guided screening

In total, 206 bacterial strains were isolated from pine forests located on Geumjeong Mountain (latitude 35.28015°, longitude 129.05062°), in Busan, South Korea. Soil samples were collected, and 1 g of each was suspended in 1 mL of a 10-mM $MgSO_4$ solution and subsequently serially diluted. From appropriate dilutions, 200 µL was smeared on nutrient agar plates and incubated for 2 days at 28°C. Single bacterial colonies with distinct shapes, colors, and morphologies were purified using the streak plate method on NA. After confirmation of pure cultures, bacterial strains were stored in 30% glycerol at −80°C until use.

For taxonomy-guided selection, bacterial genomic DNA (gDNA) was extracted using the Wizard Genomic DNA Purification Kit (Promega) following the manufacturer's instructions. The quality and concentration of the gDNA were assessed using a NanoDrop ND-2000 spectrophotometer (Thermo Fisher Scientific). For PCR amplification of the 16S rRNA sequences, the universal primers fD1 (50 -AGAGTTTGATCCTGGCTCAG -30) and rP2 (50 - ACGGCTACCTTGTTACGACTT-30) were used (52). PCR products were purified, and appropriate quality sequences were obtained and used for phylogenetic tree analysis, constructed using the maximum-likelihood method in MEGA X software (53). The 16S rRNA sequence of the reference strain *Paraburkholderia azotifigens* NF2-5-3[T] was added to the phylogenetic analysis to facilitate the selection of *Paraburkholderia*-related strains. This group includes biocontrol strains with large genomes that perform versatile functions.

Based on taxonomy-guided preliminary screening, 33 bacterial strains were selected for further analysis to identify efficient biocontrol bacterial strains against *Colletotrichum scovillei* KC05. Cultures of *C. scovillei* KC05, originally isolated from pepper plants showing anthracnose symptoms, were maintained on potato dextrose agar (PDA). To prepare fungal conidial suspensions, conidia were collected from 6 days old *C. scovillei* KC05 cultures on PDA and using sterile distilled water (SDW) containing 0.03% Tween 20. The conidial concentration of the suspension was adjusted using a hemocytometer. A dual-culture *in vitro* PDA assay was performed to select the most efficient antagonistic bacterial strain. The assay was performed by streaking a single colony from the pure culture plate of each tested bacterium as a line on the middle of the PDA plates and spotting 2 µL from conidial suspension ($10^7$ conodia/mL) of *S. scovillei* KC05 onto the two opposite sides of the plates. Control plates were inoculated with fungal conidia without bacterial streaking. Plates were incubated at 28°C and after the fungal mycelia in the control plates almost reach the middle of the plates, and the mycelial growth in the plates was evaluated. The most efficient bacterial strain was selected for further analysis and characterization.

### *In vitro* antifungal activity of the selected strain P39

The selected bacterial strain, P39, was characterized for its antifungal activity against *C. scovillei* KC05 using *in vitro* assays. The dual-culture assay on PDA was performed as described above, and mycelial growth was evaluated from photographs (in square centimeter) using ImageJ software (54).

A co-cultivation assay was performed in potato dextrose broth (PDB). Bacterial cells were harvested from an overnight culture of P39 in NB using centrifugation at 5,000 *g* at 20°C for 15 min. The cells were washed twice and resuspended in 10 mM $MgSO_4$ buffer. The bacterial concentration was adjusted using a spectrophotometer to an optical density ($OD_{600}$) of 0.6 (equivalent to ~$10^8$ CFU/mL). A conidial suspension of *C. scovillei* KC05 was prepared as described above, and its concentration was adjusted to $10^7$ conidia/mL. To perform the co-cultivation assay, 100 µL from the P39 bacterial and *C. scovillei* KC05 conidial suspensions was added to 9.8 mL of PDB and maintained in 50 mL Falcon tubes. The tubes were then incubated at 25°C with shaking for 2 days. Following incubation, fungal mycelia were collected by filtration on filter paper (Whatman #1), and the mycelial dry weight was assessed after drying for 2 days at 60°C.

To evaluate the effect of P39 cell-free culture filtrates on *C. scovillei* KC05, P39 was cultured for 1 day in PDB and filtered through 0.22 µm micro-filters (Merk Millipore, Darmstadt, Germany). From the collected cell-free suspension, 1 mL was added to 9 mL of PBD containing $10^5$ *C. scovillei* KC05 conidia/mL, prepared as explained above. Tubes were then incubated at 28°C with shaking for 2 days, and mycelial dry weights were assessed as explained above.

## *In vitro* antifungal volatiles activity of *Paraburkholderia* sp. nov. P39 against *Colletotrichum scovillei* KC05

The influence of volatile organic compounds released by P39 on *C. scovillei* KC05 was assessed *in vitro* using a bi-plate assay. The bi-plates were 90 mm Petri dishes, featuring a central partition that exclusively permitted volatiles to diffuse from one side to the other (Fisher Scientific, Pittsburgh, PA, USA). On one side of the agar plates, 2 µL from *C. scovillei* conidial suspension ($10^7$ conidia/mL) was spotted onto V8 juice agar, and on the other side, 100 µL from overnight grown P39 culture was smeared on NA. Plates smeared with sterilized NB were used as negative controls. Plates were then incubated at 28°C until the diameter of the mycelial growth in the control plates reached the central partition of the plates. The mycelial growth area was assessed using the ImageJ software, and the total produced conidia and produced conidia/$cm^2$ of fungal mycelia were counted using a hemocytometer.

## *In vivo* biocontrol activity of *Paraburkholderia* sp. nov. P39 and produced volatiles against *Colletotrichum scovillei* KC05 on green pepper

To evaluate the *in vivo* effects of strain P39 and volatiles produced by *C. scovillei* Kc05, biocontrol assays were performed using fresh green pepper (*C. annuum*) cv. Nokwang. In the dipping assay, pepper was first surface sterilized in 0.5% NaOCl for 3 min, then washed three times in SDW for 3 min, and dried on sterilized filter paper. Surface-sterilized pepper fruits were then dipped for 2 h in a bacterial suspension of strain P39 maintained in 10 mM $MgSO_4$ solution, prepared as described above. Following bacterial treatment, the pepper plants were air dried under aseptic conditions. The fungal inoculum was prepared as described above, and the conidial suspension was adjusted to $10^6$ conidia/mL. From the prepared inoculum, 2 µL was spotted into a puncture made with a sterile needle in treated pepper. After drying, the treated and inoculated peppers were maintained in surface-sterilized airtight-sealed containers with a piece of wet tissue to maintain high relative humidity. Containers were then incubated at 25°C for 5 days, and then the signs of anthracnose on inoculated peppers were photographed and measured using ImageJ software. Peppers dipped in $MgSO_4$ solution served as a negative control.

To determine the effect of volatiles produced by strain P39 on *C. scovillei* Kc05 on pepper, fruits were surface sterilized and inoculated with fungal conidia, as explained above. Inoculated pepper fruits were then kept in a square Petri plate (118 × 118 × 16.5 mm), with an NA petri plate (60 mm in diameter) smeared with 100 µL of strain P39 or $MgSO_4$ solution placed open inside the square plate. The plates were then incubated at 25°C for 5 days, and then anthracnose symptoms developed on inoculated pepper were photographed and measured using ImageJ software.

## Identification of strain P39

To identify bacterial strain P39 at the molecular level, we amplified the full-length 16S rRNA gene sequence after extracting the bacterial gDNA, as described above. The obtained sequences were compared with those from related type species using both the NCBI Basic Local Alignment Search Tool (BLAST) and phylogenetic analysis using the neighbor-joining and maximum-likelihood methods (55, 56), and the Tree topology was evaluated by conducting bootstrap analyses with 1,000 replications (57).

## Genome sequence analysis and genomic-based identification of strain P39

After analyzing the 16S rRNA gene sequence, it was not possible to identify P39 in as a specific species. Whole-genome sequencing was conducted using the bacterial gDNA extracted as described above. The quality and concentration of the obtained DNA were determined using agarose gel electrophoresis and a NanoDrop2000 spectrophotometer (Thermo Fisher Scientific, Wilmington, NC, USA). The whole P39 genome was sequenced

using the Illumina HiSeq 2500 platform, and *de novo* assembly, assembly validation, and mapping were performed at Macrogen (Seoul, South Korea).

The ANI values were computed using the OrthoANIu algorithm in EzBioCloud between the complete genome sequence of strain P39 and the closest-related strains selected based on 16S rRNA similarity and phylogenetic analysis (58). The strains included were *P. phymatum* LMG21445T, *P. fynbosensis* LMG 27177 [T], *P. dilworthii* WSM3556 [T], *P. hospita* LMG20598 [T], *P. azotifigens* NF 2-5-3 [T], and *P. bryophila* LMG 23644[T]. The genome sequences of the reference strains were retrieved from the NCBI database. dDDH values were also calculated on the genome-to-genome distance calculator web server using Formula 2 calculated at http://ggdc.dsmz.de/distcalc2.php (59). Furthermore, the P39 genome was also compared to other related genomes of *Paraburkholderia*-type strain using Type (Strain) Genome Server, and phylogenomic pairwise comparisons among the set of *Paraburkholderia* genomes were conducted using the genome blast distance phylogeny and accurate intergenomic distances inferred under the algorithm "trimming" and distance formula $d_5$ with 100 distance replicates (59, 60). Based on these intergenomic distances, a balanced minimum evolution tree was constructed using FASTME 2.1.6.1 with branch support inferred from 100 pseudo-bootstrap replicates each (61). To cluster the genomes into species, type-based species clustering was performed using a 70% dDDH radius around each of the used type strains (60). The tree was visualized using iTOL v5 (62).

The genome sequence of the P39 strain was annotated by RAST version 2.0 and the SEED Viewer server (63). Further annotation was performed using Prokka v1.12 (64). Finally, the antiSMASH tool was used to predict secondary metabolite biosynthesis gene clusters (65).

## Mycophagy assay of *Paraburkholderia busanensis* P39 against *Colletotrichum scovillei* KC05

To investigate whether the selected strain P39 can consume *C. scovillei* KC05 fungal mass and increase its population, a mycophagy assay was designed. Mycelial mass of *C. scovillei* KC05 was obtained from 2-day cultures in PDB at 25°C with shaking. Mycelia were collected aseptically, washed twice in SDW to eliminate fungal metabolites, and weighed after excess water was removed. Bacterial suspensions from overnight cultures were prepared as described previously, and the concentration was adjusted to an $OD_{600}$ of 0.6.

The assay was performed with normal NB and on NB diluted to 10×, 100×, and 1,000× with SDW. The assay was divided into three sets of tubes [1, (+) fungi (+) bacteria; 2, (+) fungi (−) bacteria; 3, (−) fungi (+) bacteria] with three replicates each, with a total volume of 3 mL. Each tube contained 6 mg *C. scovillei* KC05 mycelia and 100 µL of the prepared bacterial suspension. Control tubes consisted of untreated tubes with or without the bacterial suspension; a 10-mM $MgSO_4$ solution was used as the negative control for bacterial treatment, whereas SDW was used as the negative control for fungal mycelial treatment.

The tubes were incubated for 1 day at 28°C with shaking. After incubation, the fungal mycelia were collected on sterilized filter paper (Whatman #1), and weighed after excess water was removed. The bacterial population of strain P39 was evaluated by smearing 200 µL onto NA plates after serial dilutions, and the CFUs were counted after 2 days of incubation at 28°C. In another set of assays, the interactions between strain P39 and *C. scovillei* KC05 were investigated using an Optinity KB-600F microscope equipped with CoolLED pE300-WHITE illumination (CoolLED). Images were acquired using a KCS3-23S CMOS camera (Korea LabTech, Seoul, Korea).

## Characterization of *Paraburkholderia busanensis* sp. nov. P39

Cellular fatty acids, polar lipids, and quinones were analyzed at the KCCM services (Seoul, South Korea) following the standard protocol (http://www.kccm.or.kr/). Briefly, cellular fatty acids were extracted using the standard MIDI protocol (Sherlock Microbial

Identification System version 6.3) and analyzed by gas chromatography (Agilent Technologies, 6850). The polar lipid content was analyzed by two-dimensional thin-layer chromatography with different staining solutions, including ninhydrin, molybdophosphoric acid, and -naphthol, sprayed on the plate to detect different types of lipids. Finally, high-performance liquid chromatography was used to identify the extracted quinones following standard methods.

To further characterize the biochemical properties and enzymatic activities of strain P39, commercially available API 20NE and API Zym kits (bioMérieux, France) were used according to the manufacturer's instructions (https://www.biomerieux-diagnostics.com/). The oxidase activity of the isolates was tested using a 1% tetramethyl-p-phenylenediamine dihydrochloride solution. To assess bacterial growth under different conditions, optical density measurements were taken at different incubation temperatures (4–42°C, with 2–4°C increments), pH levels (4–9, adjusted using an appropriate buffer with 0.5 increments), and NaCl concentrations (0%–2.5%, adjusted with 0.5% increments) in R2A liquid media.

Other properties related to biocontrol and antagonistic activities, including cellulase, siderophore production, phosphate solubilization, and protease activities, were evaluated *in vitro* using agar plate assays. Cellulase activity was tested on carboxymethyl cellulose medium, and siderophore production was tested on chrome azurol S agar assay, phosphate solubilization was tested on Pikovskaya's medium, and phosphate solubilization and extracellular protease activity were tested on skim milk agar medium. The media and constituents were prepared as previously described (66, 67).

Chitinase activity was assessed using the enzyme assay kit (Sigma-Aldrich, St. Louis, MO, USA, Cat. No. CS0980) following the manufacturer's instructions. The assay measured both exo- and endo-chitinolytic activity using a colorimetric method with three different substrates: 4-nitrophenyl-N-acetyl-β-d-glucosaminide, 4-nitrophenyl-β-d-N,N′,N″-triacetylchitotriose, and 4-nitrophenyl-N,N′-diacetyl-β-d-chitobioside. A positive control was included, which consisted of pure chitinase enzyme from *Trichoderma viride* provided as a component in the kit. The absorbance, which is proportional to the amount of released *p*-nitrophenol, was measured using a Bio-Rad microplate reader, model 680 (BioRad, Hercules, CA, USA), and the calculations were made according to the manufacturer's instructions as one unit of the enzyme equals the generation of 1 µmol of *p*-nitrophenol/min from the used substrate. The kit was used for assessment of endo-chitinase activity in the presence or absence of *C. scovillei* KC05 fungal mycelia on NB prepared as explained above.

## Statistical analysis

The experiments were conducted twice with three replicates per treatment. For the analysis of microbial populations, the data were subjected to log transformation. Fungal mycelial growth and anthracnose lesion area were measured using ImageJ software (54). Statistical analysis was performed using the general linear model procedures, with analysis of variance being conducted. Post hoc comparisons were made using the least significant difference test at a significance level of $P < 0.05$. All statistical analyses were carried out using SAS software (SAS Institute, Cary, NC).

## ACKNOWLEDGMENTS

This research was supported by the Basic Science Research Program of the National Research Foundation (NRF), funded by the Ministry of Education (2022R1A2C1003190), Republic of Korea.

The authors declare that they have no competing financial interests or personal relationships that may have influenced the work reported in this study.

## AUTHOR AFFILIATIONS

[1]Department of Microbiology, Pusan National University, Busan, South Korea

[2]Department of Integrated Biological Science, Pusan National University, Busan, South Korea

[3]Department of Plant Pathology, Faculty of Agriculture, Cairo University, Giza, Egypt

## AUTHOR ORCIDs

Mohamed Mannaa  http://orcid.org/0000-0002-9079-8970

Young-Su Seo  http://orcid.org/0000-0001-9191-1405

## FUNDING

| Funder | Grant(s) | Author(s) |
|---|---|---|
| National Research Foundation of Korea (NRF) | 2022R1A2C1003190 | Young-Su Seo |

## AUTHOR CONTRIBUTIONS

Mohamed Mannaa, Conceptualization, Data curation, Investigation, Methodology, Visualization, Writing – original draft | Gil Han, Investigation, Methodology, Validation, Visualization | Taeho Jeong, Formal analysis, Methodology | Minhee Kang, Methodology, Validation | Duyoung Lee, Methodology, Validation | Hyejung Jung, Data curation, Software | Young-Su Seo, Conceptualization, Funding acquisition, Project administration, Resources, Supervision, Validation, Writing – review and editing

## DATA AVAILABILITY

The genome sequence of *P. busanensis* P39 sp. nov. has been deposited in the National Center for Biotechnology Information (NCBI) under the BioProject accession no. PRJNA638731 and BioSample SAMN15207046.

## ADDITIONAL FILES

The following material is available online.

### Supplemental Material

**Fig. S1 (Spectrum02426-23-s0001.pdf).** Cellulase activity was determined on CMC medium, phosphate solubilization on Pikovskaya's medium, siderophore production on CAS medium, and extracellular protease activity on skim milk medium.
**Table S1 (Spectrum02426-23-s0002.docx).** List of the isolated strains and the accession numbers of their 16S rRNA partial sequences on NCBI GenBank database.

### Open Peer Review

**PEER REVIEW HISTORY (review-history.pdf).** An accounting of the reviewer comments and feedback.

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
