## [Reviewer comments · Microbiology Spectrum]

Microbiology Spectrum

Taxonomy-guided selection of *Paraburkholderia busanensis* sp. nov.: A versatile biocontrol agent with mycophagy against *Colletotrichum scovillei* causing pepper anthracnose

Mohamed Mannaa, Gil Han, Taeho Jeong, Minhee Kang, Dooyoung Lee, Hyejung Jung, and Young-Su Seo

Corresponding Author(s): Young-Su Seo, Pusan National University

Review Timeline:

Submission Date:	June 9, 2023
Editorial Decision:	July 11, 2023
Revision Received:	July 21, 2023
Editorial Decision:	August 26, 2023
Revision Received:	August 27, 2023
Accepted:	September 4, 2023

Editor: Renee Arias

Reviewer(s): Disclosure of reviewer identity is with reference to reviewer comments included in decision letter(s). The following individuals involved in review of your submission have agreed to reveal their identity: Saul Fraire-Velázquez (Reviewer #1); Angel Andrade Torres (Reviewer #4)

Transaction Report:

DOI: <https://doi.org/10.1128/spectrum.02426-23>

July 11, 2023

Prof. Young-Su Seo
Pusan National University
Department of Microbiology
Busan
Korea (South), Republic of

Re: Spectrum02426-23 (Taxonomy-guided selection of *Paraburkholderia busanensis* sp. nov.: A versatile biocontrol agent with mycophagy against *Colletotrichum scovillei* causing pepper anthracnose)

Dear Prof. Young-Su Seo:

Link Not Available

Sincerely,

Renee Arias

Journals Department
Reviewer comments:

Reviewer #1 (Comments for the Author):

The text in lines 122-125 sounds repetitive with lines 128-131. Figure 1B and Fig 2B refer to the same thing although they do not fully agree. Need to be specified or corrected.

Highlight some of the specific genetic characteristics for each chromosome. For example, size, %GC, CDS, cluster of biosynthetic genes, etc.

In lines 222-225, this point deserves further analysis and discussion of why in the presence of mycelium of the fungus,

endochitinase activity decreases, compared to treatment where no mycelium was added.

In line 260, 100X is repeated, correct.

In line 415, what means fragments ?

In line 440, more and more recent bibliographic citations can be added.

In lines 441-443, specify a bit about species and mechanisms.

Reviewer #3 (Comments for the Author):

I commend the authors of this research article for executing a study with such depth, utilizing a number of different techniques for novel biocontrol agent selection and characterization. The characterization of the biocontrol agent is thorough and unveils several important metrics for understanding antagonistic activity against the target pathogen and disease. I especially appreciated the development of a potential mechanism of action for chitin utilization. I couldn't find any major issues with the methods or conclusions, but do have some minor comments and suggested changes, below.

Lines 83-84: "Among the examples of registered products based on biocontrol agents"
This is not a complete sentence.

Line 150: "dipping in a fresh pepper bacterial suspension"
Suggest changing to "dipping a fresh pepper in a bacterial suspension".

Line 362: "To conduct taxonomy-guided screening"
Suggest changing to "To conduct the taxonomy-guided screening"

Lines 576-577: "After analyzing the 16S rRNA gene sequence, it was not possible to identify P39 in a specific species."
Suggest changing to "as a specific species"

Lines 644-646: pH and NaCl increments are mentioned, but temperature increments are not.

Table 1: "Acession" should be "Accession"

Table 4:
"Root nodule", "root nodule", inconsistent capitalization of "root".
"+ W" not used in table but mentioned in caption.
Lines 890-891: "+ W, weakly positive; + W, weak positive" mentioned twice in caption.
Lines 890-891: "Positive... negative" different capitalization used in caption.

Line 934: "and incubated in on sealed square dish"
Suggest changing to "and incubated in a sealed square dish"

Line 970: "P. busanensis" should be italic.

Line 981: "circular map of the P. busanensis P39 genome."
the word "the" should not be italic.

Lines 1022-1023: "showing intact mycelia and fungal undepleted cellular contents without cell wall damage"
Suggest changing to "showing intact mycelia with undepleted cellular contents and without cell wall damage"

Line 1025: "rupture in the mycelial"
Suggest changing to "rupture in the mycelia"

Line 1027: "The red arrows indicating points of rupturing fungal cell wall"
Suggest changing to "The red arrows indicate points of fungal cell wall rupture"

Staff Comments:

Preparing Revision Guidelines

Please return the manuscript within 60 days; if you cannot complete the modification within this time period, please contact me. If you do not wish to modify the manuscript and prefer to submit it to another journal, please notify me of your decision immediately so that the manuscript may be formally withdrawn from consideration by Microbiology Spectrum.

[Response letter]

21-July-2023

Dear Editor,

Subject: Submission of revised paper [Manuscript Number: Spectrum02426-23]

Thank you for your email dated 22-Feb-2023 enclosing the reviewers' comments. We are sincerely grateful to both you and the reviewers for the insightful comments and suggestions for improvement. We have taken these recommendations seriously and have revised our manuscript accordingly. We are pleased to enclose the revised version of our manuscript titled "**Taxonomy-guided selection of *Paraburkholderia busanensis* sp. nov.: A versatile biocontrol agent with mycophagy against *Colletotrichum scovillei* causing pepper anthracnose**". We are confident that the manuscript has significantly improved after the revisions.

In response to the comments, we have addressed each of the points raised, and have included the necessary changes in the accompanying revised manuscript. To ensure transparency, all changes were made using the track changes option in word processing. We appreciate the thorough review process and are grateful for the opportunity to improve our work based on the reviewers' feedback.

Corresponding author,

Reviewer Comments:

Reviewer 1:

(Comments for the Author):

The text in lines 122-125 sounds repetitive with lines 128-131. Figure 1B and Fig 2B refer to the same thing although they do not fully agree. Need to be specified or corrected.

>> We thank the reviewer for dedicating time to thoroughly review our manuscript and for offering valuable suggestions. In response to the reviewer's feedback, we would like to clarify the methodology and address potential points of confusion.

Initially, we performed a preliminary screening dual-culture assay, evaluating a total of 33 *Paraburkholderia*-related strains. Through this screening, we identified strain P39 (as depicted in Figure 1B) as the most promising candidate. Subsequently, strain P39 was selected for further characterization, which included an additional dual-culture assay shown in Figure 2A and B.

To enhance clarity and eliminate potential ambiguity, we have explicitly stated that the results presented in Figure 1B correspond to the outcomes from the preliminary screening assay. By doing so, we aim to ensure that readers can discern the two separate stages of our investigation. The sentence in L 123-124) was fixed for clarity.

Highlight some of the specific genetic characteristics for each chromosome. For example, size, %GC, CDS, cluster of biosynthetic genes, etc.

>> As suggested, we have added supplementary table 2 containing the specific features and genetic characteristics for each chromosome.

In lines 222-225, this point deserves further analysis and discussion of why in the presence of mycelium of the fungus, endochitinase activity decreases, compared to treatment where no mycelium was added.

>> Thank you for your insightful comments. Concerning the observed slight decrease in the endochitinase activity, we propose that this reduction can be primarily attributed to the utilization of a portion of the enzyme activity during the metabolic processes involved in the degradation of

the fungal mycelial cell wall, which contains chitin. Our mycophagy assay confirmed a decrease in the fungal mycelial weight in the presence of bacteria, and it is well-established that the utilization of chitin is a crucial aspect of this process.

In response to your recommendation, we have taken the opportunity to provide a more detailed explanation in our revised manuscript (see Line 225-227) to clarify this point.

In line 260, 100X is repeated, correct.

>> The typo has been rectified to 100X and 1000X (L262)

In line 415, what means fragments ?

>> The typo has been rectified from “fragments” to “fragin”, the name of the secondary metabolite (L415)

In line 440, more and more recent bibliographic citations can be added.

>> As per the reviewer's suggestion, we have incorporated more recent bibliographic citations that include additional similar observations (L 438 to 460).

In lines 441-443, specify a bit about species and mechanisms.

>> Based on the reviewer's feedback, we have made improvements to the relevant section. We have included additional details about the specific species known to display mycophagy, and we have expanded upon the mechanisms and strategies employed by the selected species in the mycophagy activities (L 438 to 460). Moreover, we have incorporated relevant citations to support these additions.

Reviewer 2:

(Comments for the Author):

I commend the authors of this research article for executing a study with such depth, utilizing a number of different techniques for novel biocontrol agent selection and characterization. The characterization of the biocontrol agent is thorough and unveils several important metrics for understanding antagonistic activity against the target pathogen and disease. I especially appreciated the development of a potential mechanism of action for chitin utilization. I couldn't find any major issues with the methods or conclusions, but do have some minor comments and suggested changes, below.

>> We thank the reviewer for the positive and encouraging feedback on our research article. We appreciate your commendation of the depth of our study, the use of various techniques for biocontrol agent selection, and the thorough characterization provided. Your recognition of the potential mechanism of action for chitin utilization is truly motivating. We have carefully considered and addressed your valuable suggestions to enhance the manuscript further.

Lines 83-84: "Among the examples of registered products based on biocontrol agents"

This is not a complete sentence.

>> Thank you for bringing this to our attention. We have addressed the issue by completing the sentence and including the example of registered products.

Line 150: "dipping in a fresh pepper bacterial suspension"

Suggest changing to "dipping a fresh pepper in a bacterial suspension".

>> The sentence was corrected as recommended.

Line 362: "To conduct taxonomy-guided screening"

Suggest changing to "To conduct the taxonomy-guided screening"

>> The sentence was corrected as recommended.

Lines 576-577: "After analyzing the 16S rRNA gene sequence, it was not possible to identify P39 in a specific species."

Suggest changing to "as a specific species"

>> The sentence was corrected as recommended.

Lines 644-646: pH and NaCl increments are mentioned, but temperature increments are not.

>> Temperature increments were mentioned as recommended.

Table 1: "Acession" should be "Accession"

>> The pointed typo was corrected as recommended.

Table 4:

"Root nodule", "root nodule", inconsistent capitalization of "root".

>> The pointed typo was corrected.

"+ W" not used in table but mentioned in caption.

>> The pointed error was corrected.

Lines 890-891: "+ W, weakly positive; + W, weak positive" mentioned twice in caption.

>> The pointed error was corrected.

Lines 890-891: "Positive... negative" different capitalization used in caption.

>> The pointed typo was corrected.

Line 934: "and incubated in on sealed square dish"

Suggest changing to "and incubated in a sealed square dish"

>> The pointed typo was corrected.

Line 970: "*P. busanensis*" should be italic.

>> The pointed typo was corrected.

Line 981: "circular map of the *P. busanensis* P39 genome."

the word "the" should not be italic.

>> The pointed typo was corrected.

Lines 1022-1023: "showing intact mycelia and fungal undepleted cellular contents without cell wall damage"

Suggest changing to "showing intact mycelia with undepleted cellular contents and without cell wall damage"

>> The sentence was corrected as suggested.

Line 1025: "rupture in the mycelial"

Suggest changing to "rupture in the mycelia"

>> The pointed typo was corrected.

Line 1027: "The red arrows indicating points of rupturing fungal cell wall"

Suggest changing to "The red arrows indicate points of fungal cell wall rupture"

>> The sentence was corrected as suggested.

August 26, 2023

Prof. Young-Su Seo
Pusan National University
Department of Microbiology
Busan
Korea (South), Republic of

Re: Spectrum02426-23R1 (Taxonomy-guided selection of *Paraburkholderia busanensis* sp. nov.: A versatile biocontrol agent with mycophagy against *Colletotrichum scovillei* causing pepper anthracnose)

Dear Prof. Young-Su Seo:

Thank you for submitting your manuscript to Microbiology Spectrum. Before we can accept your manuscript for publication, there are few more changes recommended by the reviewers, those changes are accompanying this letter. When submitting the revised version of your paper, please provide (1) point-by-point responses to the issues raised by the reviewers as file type "Response to Reviewers," not in your cover letter, and (2) a PDF file that indicates the changes from the original submission (by highlighting or underlining the changes) as file type "Marked Up Manuscript - For Review Only". Please use this link to submit your revised manuscript - we strongly recommend that you submit your paper within the next 60 days or reach out to me. Detailed instructions on submitting your revised paper are below.

Link Not Available

Sincerely,

Renee Arias

Journals Department
Reviewer comments:

Reviewer #3 (Comments for the Author):

Thank you for addressing the previous comments. Upon further inspection, I found several minor misspellings that should be addressed.

From the line numbers of the PDF:

Line 24 "Paraburkholderi" misspelled

Line 199 and 216 "busanesnsis" misspelled

Line 374 "Paraburkholderia" misspelled

Lines 688, 967, 972 "imageJ" should be ImageJ

Line 1030 "repression" misspelled

Reviewer #4 (Comments for the Author):

I acknowledge the authors for their well-accomplished work regarding the identification and characterization of a new species of Paraburkholderia with potential as biocontrol agent. This is a comprehensive study contributing to the understanding of bacterial antagonism against fungal pathogens. I have some minor issues and suggestions.

Line 113, please include some info regarding site of bacterial pool isolation

Line 131, "co-cultivation" is duplicated

Line 141 and 400, I will argue that there is not a clear direct effect of volatiles over conidiation of *C. scovillei* (Fig. 3C), it seems that lesser conidia count only reflects fungal growth inhibition.

Please refer Table 3 at the beginning of description of corresponding data (line 199-200)

Line 413, I couldn't find any reference in the manuscript or tables regarding the "various types of secretion system". It would be nice to indicate which kind of secretion systems are encoded by *P. busanensis* P39, particularly if there is a T3SS.

Figure 7. I find this figure unnecessary. Data in Fig. 7A is better summarized in Table S2. In fact, data in Table S2 can be merged with Table 2. And for Fig. 7B, with only 1642 (22%) of CDSs categorized by RAST software, subsystem categorization is probably unreliable by not considering over 3000 CDS with functional assignments (according to Table 2).

Staff Comments:

Preparing Revision Guidelines

Please return the manuscript within 60 days; if you cannot complete the modification within this time period, please contact me. If you do not wish to modify the manuscript and prefer to submit it to another journal, please notify me of your decision immediately so that the manuscript may be formally withdrawn from consideration by Microbiology Spectrum.

Thank you for addressing the previous comments. Upon further inspection, I found several minor misspellings that should be addressed.

From the line numbers of the PDF:

Line 24 "Paraburkholderi" misspelled

Line 199 and 216 "busanesnsis" misspelled

Line 374 "Paraburkhoderia" misspelled

Lines 688, 967, 972 "imageJ" should be ImageJ

Line 1030 "repressession" misspelled

[Response letter]

28-August-2023

Dear Editor,

Subject: Submission of revised paper [Manuscript Number: Spectrum02426-23] – R2

Thank you for your email from 27-August-2023 sharing the reviewers' feedback for our manuscript's second revision. We deeply appreciate the constructive suggestions and have updated our manuscript titled "**Taxonomy-guided selection of *Paraburkholderia busanensis* sp. nov.: A versatile biocontrol agent with mycophagy against *Colletotrichum scovillei* causing pepper anthracnose**" in response. We believe these revisions enhance its quality considerably.

To address all concerns transparently, we've documented every change using the 'track changes' feature in our word processor. We value the meticulous review process and thank you for this chance to refine our work based on expert feedback.

Corresponding author,

Reviewer Comments:

Reviewer 3:

(Comments for the Author):

Thank you for addressing the previous comments. Upon further inspection, I found several minor misspellings that should be addressed.

>>> Thank you for your time and keen observations. We've benefited greatly from your feedback and will promptly address the misspellings. Your help in refining our manuscript is truly appreciated.

From the line numbers of the PDF:

Line 24 "Paraburkholderi" misspelled

>>> The typo was corrected in the revised version (L24).

Line 199 and 216 "busanesnsis" misspelled

>>> We are so thankful for noticing this important one!. The typo was corrected in the revised version (L197 and 214).

Line 374 "Paraburkhoderia" misspelled

>>> The typo was corrected in the revised version (L370).

Lines 688, 967, 972 "imageJ" should be ImageJ

>>> The typo was corrected in the revised version (L682, 957, 961).

Line 1030 "repression" misspelled.

>>> The typo was corrected in the revised version (L1011).

Reviewer #4

(Comments for the Author):

I acknowledge the authors for their well-accomplished work regarding the identification and characterization of a new species of Paraburkholderia with potential as biocontrol agent. This is a comprehensive study contributing to the understanding of bacterial antagonism against fungal pathogens. I have some minor issues and suggestions.

>>> We sincerely thank the reviewer for the constructive feedback and kind recognition of our work. Your insightful comments and suggestions are invaluable, and we are eager to address the minor issues pointed out.

Line 113, please include some info regarding site of bacterial pool isolation

>>> We've added details about the site of bacterial pool isolation, specifying it as Geumjeong Mountain (latitude 35.28015°, longitude 129.05062°) in Busan, South Korea L(491-492) in the revised version.

Line 131, "co-cultivation" is duplicated

>>> We have corrected the sentence (L130) in the revised version.

Line 141 and 400, I will argue that there is not a clear direct effect of volatiles over conidiation of *C. scovillei* (Fig. 3C), it seems that lesser conidia count only reflects fungal growth inhibition.

>>> We appreciate the reviewer's perspective on the influence of bacterial volatiles on mycelial growth, and how this might inadvertently impact conidia counts. To address this, it is vital to clarify our methodology: rather than merely counting total conidia produced, we also normalized these counts to the mycelial growth area, providing a measure of conidia produced per cm² of fungal growth. This was done by dividing the total conidia count by the mycelial growth area in each plate.

In both metrics (total conidia and conidia per cm²) the bacterial volatiles showed a significant reduction in conidia production. This is visualized in Figure 3C: the plain dark blue bars represent total conidia, while the light blue bars with stripes indicate conidia per cm².

Please refer Table 3 at the beginning of description of corresponding data (line 199-200)

>>> As suggested we have referred to Table 3 at the beginning of the description (L198) in the revised version.

Line 413, I couldn't find any reference in the manuscript or tables regarding the "various types of secretion system". It would be nice to indicate which kind of secretion systems are encoded by *P. busanensis* P39, particularly if there is a T3SS.

>>> Thank you for pointing out the oversight. Based on our initial annotations, we identified that *P. busanensis* P39 indeed encodes for a range of membrane transport within the subsystems, including type secretion systems, specifically Type II, Type III, and Type VI secretion systems,. We are in the process of further investigating the role of these secretion systems in the bacterium, especially their potential implications in biocontrol activity and bacterial-fungal interactions. Our findings will be detailed in an upcoming manuscript dedicated to this topic. Meanwhile, we have updated the current manuscript to provide more clarity on this aspect. We added the membrane transport to the annotation of subsystems (L196-197) and added further details in the discussion (L411-415) in the revised version.

Figure 7. I find this figure unnecessary. Data in Fig.7A is better summarized in Table S2. In fact, data in Table S2 can be merged with Table 2. And for Fig. 7B, with only 1642 (22%) of CDSs categorized by RAST software, subsystem categorization is probably unreliable by not considering over 3000 CDS with functional assignments (according to Table 2).

>>> We agree with the reviewer's comment that Figure 7 might be unnecessary. Based on the reviewer's suggestion, we have removed Figure 7 and Table S2, consolidating the genome features in Table 2.

We wish to emphasize that while RAST assigns genes into subsystems (sets of interrelated functional roles), functional assignments denote specific gene functions. Therefore, not every gene with a functional assignment aligns with a RAST-defined subsystem. Indeed, a majority of genes might operate outside these established subsystems. Nevertheless, the 22% categorized by RAST encapsulates essential biological processes and complexes, serving as a complement to the more extensive functional assignment data.

September 4, 2023

Prof. Young-Su Seo
Pusan National University
Department of Microbiology
Busan
Korea (South), Republic of

Re: Spectrum02426-23R2 (Taxonomy-guided selection of *Paraburkholderia busanensis* sp. nov.: A versatile biocontrol agent with mycophagy against *Colletotrichum scovillei* causing pepper anthracnose)

Dear Prof. Young-Su Seo:

Your manuscript has been accepted, and I am forwarding it to the ASM Journals Department for publication. You will be notified when your proofs are ready to be viewed.

Sincerely,

Renee Arias
Editor, Microbiology Spectrum
